# Organoids: Construction and Application in Gastric Cancer

**DOI:** 10.3390/biom13050875

**Published:** 2023-05-22

**Authors:** Chengdong Huo, Xiaoxia Zhang, Yanmei Gu, Daijun Wang, Shining Zhang, Tao Liu, Yumin Li, Wenting He

**Affiliations:** 1Department of the Second Clinical Medical College, Lanzhou University, Lanzhou 730030, China; 2Key Laboratory of Digestive System Tumors of Gansu Province, Lanzhou 730030, China; 3Department of Ophthalmology, Lanzhou University Second Hospital, Lanzhou 730030, China

**Keywords:** organoid, tumor organoid, stem cell, 3D cell culture, patient-derived organoids (PDO)

## Abstract

Gastric organoids are biological models constructed in vitro using stem cell culture and 3D cell culture techniques, which are the latest research hotspots. The proliferation of stem cells in vitro is the key to gastric organoid models, making the cell subsets within the models more similar to in vivo tissues. Meanwhile, the 3D culture technology also provides a more suitable microenvironment for the cells. Therefore, the gastric organoid models can largely restore the growth condition of cells in terms of morphology and function in vivo. As the most classic organoid models, patient-derived organoids use the patient’s own tissues for in vitro culture. This kind of model is responsive to the ‘disease information’ of a specific patient and has great effect on evaluating the strategies of individualized treatment. Herein, we review the current literature on the establishment of organoid cultures, and also explore organoid translational applications.

## 1. Introduction

Gastric cancer is the fifth most common malignant tumor, with over one million people diagnosed each year worldwide. Although the incidence and mortality rates for gastric cancer have declined globally over the past 50 years, it remains the third leading cause of cancer deaths [1]. Perioperative chemotherapy supported by endoscopic treatment or surgery is the only curative treatment available. However, early diagnosis of gastric cancer is still difficult because of the lack of reliable biomarkers. Most preclinical research of gastric cancer is reliant on cell lines and animal models, but both fail to reproduce the original human tumors. Organoid models are cell populations obtained by relying on three-dimensional (3D) cell culture techniques and stem cell proliferation in vitro, with biological characteristics highly similar to those in the tissue where they originate from. They can also maintain their ability to self-renew and proliferate in vitro. Patient-derived organoids (PDO) have been known as a better biological model that can more directly reflect the specific functions, developmental conditions, and disease characteristics of patient-derived tissues. The gastric cancer organoids are a kind of biological tumor model constructed using tumor tissue based on the above-mentioned culture model. In recent years, gastric organoids and gastric cancer organoid models have also been successfully developed, opening up new possibilities for a variety of clinical translational applications. Both have significant advantages over traditional biological models, but there are still some differences in the physiological functions and construction methods of different organoid models. In this review, we will illustrate the characteristics of gastric organoid models from different stem cell sources, the different methods of the organoids’ construction, and the translational applications of gastric organoids.

## 2. Organoids Definition and Characteristics

Smith and Cochrane firstly used ‘organoids’ to describe a cystic teratoma in the pararenal tissue of an infant in 1946 [2]. Since then, researchers have used the term to describe tissues or tumors with self-organization during cell sorting and reaggregation. Dutch scientist Hans Clevers successfully constructed small intestine organoids in vitro using mouse LGR5^+^ small intestine stem cells for the first time [3]. Subsequently, liver, pancreas, brain and retina organoids were constructed, which also redefined the concept of organoids [4]. It is generally accepted that organoids are cell populations obtained through proliferation of stem cells under a 3D cell culture condition in vitro. Organoids always have the ability to self-assemble and self-renew. In addition, organoids are highly similar in cellular composition and function to tissues in vivo [2]. With the continuous advancement of the studies of organoids, Sylvia F Boj et al. successfully established the first pancreatic tumor organoid model in 2015 [5]. They introduced using tumor tissue as the source for tumor organoid models for the first time. In the following years, different types of tumor organoid models were successfully constructed, such as esophageal, gastric, colorectal, lung, bladder and endometrial cancers [6,7,8]. Tumor organoid models can perform most of the biological characteristics of tumor cells in vitro, becoming more ‘accurate’ biological models for basic tumor studies, such as immunotherapy, new drug development and drug screening. Although it has only been a few years since the successful construction of the gastric organoids and gastric tumor organoids, there are a growing number of studies on stomach-related diseases using such models.

## 3. Construction of Organoids

### 3.1. Cells in Organoids Culture

The construction of organoid models is based on stem cell culture techniques. Gastric cancer organoids can be derived from adult stem cells (ASC), induced pluripotent stem cells (iPSC), and embryonic stem cells (ESC). However, there are ethical restrictions on ESC research, so it will not be further addressed in this review. ASC and iPSC-derived gastric organoids are currently the two dominant culture methods. Both essentially utilize the proliferation and differentiation abilities of stem cells, but they remain somewhat different in terms of model construction and biological characteristics (Table 1) as follows: (1) in terms of cell composition, ASC-derived gastric organoids contain only epithelial-derived cells (e.g., principal cells, mural cells), whereas iPSC are able to differentiate both epithelial-derived cells and various stromal cells (e.g., smooth muscle cells, fibroblasts); (2) in terms of proliferative capacity, although both can undergo multiple passages in vitro, ASC-derived gastric organoids are likely to have a greater proliferative capacity; (3) in terms of modelling time, ASC-derived gastric organoids need less time to construct, usually only 1–2 weeks, to obtain more stable models, compared to 1–2 months for iPSC-derived models; (4) in terms of histomorphology, ASC-derived gastric organoids resemble adult tissues, while iPSC-derived are more similar to embryonic tissues [9].

Human-derived tumor organoids are also known as patient-derived organoids (PDO). Depending on the method of acquisition, PDO are divided into tissue-derived, liquid biopsy-derived, and enriched stem cell-derived. First, patient tissue derived is the most common method of acquisition. These tissues obtained by surgery or endoscopic biopsy are decomposed by relevant enzymes to extract tumor cells for the cultivation in vitro [10]. The tumor organoids are relatively easy to construct and have the highest similarity to the tumor in the patients, but the process of obtaining the tissues is often invasive. Second, patient fluid biopsy has been an emerging test in recent years. Circulating tumor cells (CTCs) are obtained by screening and analyzing the patient’s blood, ascites and even urine, and then used to construct tumor organoids. The advantages of these tumor organoids are the non-invasive process and the possibility of dynamic monitoring at different stages of the tumor, but it is relatively difficult to construct [11,12]. Third, tumor stem cells are a population of cells within the tumor tissue with the ability to proliferate and differentiate, and they play an important function in tumorigenesis, invasion and metastasis; they are also a major factor in the construction of tumor organoids. It has been found that tumor stem cells can be isolated by using flow cytometry or other technical methods for cell culture. These tumor organoids have been shown to restore the heterogeneity of tumors [13,14]. As adult stem cells are present in the gastric glands, surgical and biopsy-obtained tissue blocks often require isolation of glands. The methods used to isolate the gland include physical pressure and enzymatic reaction. Glands in the sample can be isolated by gently pressuring the tissue using a slide. This method can better maintain the activity of the cells but is less efficient [15]. Enzymatic reaction is the most common method of isolation, using various enzymes to break the connections between cells. This method is dependent on the type and activity of enzymes. Currently, there are different methods and enzymes used to isolate glands in many studies of gastric PDO. We summarized the studies on this issue within the last 5 years (Table 2).

### 3.2. Organoid Culture Systems

#### 3.2.1. Culture Media

The basal medium provides glucose, non-essential amino acids and other substances necessary for cell growth. The organoids culture media contain not only the basal media, but also cytokines, small molecule inhibitors and hormones [33,34]. Moreover, B27, *L*-glutamine and ascorbic acid are also essential components of the basal medium [35,36]. Cytokines are crucial components of culture media that help to maintain the activity of stem cells. It was found that the Wnt signaling pathway is an essential pathway for stem cell proliferation and differentiation. Wnt-3A, a family member of this signaling pathway, is also one of the substantial cytokines within the organoids medium [37,38]. *R*-spondin-1 inhibits the RNF43/ZNRF3 signaling pathway and, in maintaining continuous activation of the Wnt signaling pathway by binding to Lgr5, is therefore another essential cytokine for maintaining stem cell activity in medium. Furthermore, Lgr5 is also considered an important marker of stem cells [39,40,41]. In addition, Noggin, epidermal growth factor (EGF), fibroblast growth factor (FGF) and vascular endothelial growth factor (VEGF) also play a role in the culture of organoids and need to be adapted to the requirements of different organoid models [42,43,44,45,46]. As cofactors of the above cytokines, small molecule inhibitors, such as CHIR99021, Y-27632, A83-01 and SB-431542, can assist cytokines in maintaining stem cell activity and reducing apoptosis [47,48,49,50,51,52,53]. Hormones are necessary for specific types of organoids. For example, gastrin effectively promotes the proliferation and differentiation of mouse gastric organoids via cholecystokinin 2 receptor (CCK2R) [54].

#### 3.2.2. Extracellular Matrix Materials

The traditional two-dimensional (2D) cell culture method has been used for over a hundred years and is still the most common cell culture method. However, with the continuous advancement of research, the 2D cell culture model could no longer fulfill the scientific research, so the 3D cell culture model has emerged. Compared with the traditional culture mode, the 3D cell culture model can retain the biological characteristics of tissues in vivo.

Three-dimensional cell culture techniques are based on the application and development of various matrix materials, which can be divided into two major types, natural matrix and synthetic matrix, depending on the source of the matrix materials. Matrigel is the most commonly used type of natural matrix, which is derived from basement membrane proteins secreted by Engelbreth–Holm–Swarm (EHS) mouse chondrosarcoma cells and is very similar to human basement membrane components. It can effectively support cells to maintain 3D structures while promoting the proliferation and differentiation of stem cells to form specific organoids [55]. Synthetic matrices are a type of polymeric material obtained through bioengineering, with the advantages of a porous structure and stable mechanical properties. It can be constructed artificially on an extracellular matrix and has more plasticity compared with a natural matrix. Various artificial matrices have been synthesized for organoids culture, such as polycaprolactone porous scaffolds. Currently, researchers have developed ‘composite’ matrix material, which owns two types of extracellular matrix. This material is obtained by implanting fibroblasts inside the synthetic scaffolds, using the fibroblasts to produce a natural matrix, and removing the fibroblasts afterwards. ‘Composite’ matrix material is more conducive to the generation of organoids and has a much shorter modeling time compared to one matrix alone [56,57].

#### 3.2.3. Organoid Culture Modes

With the advancement of cell culture technology, there are more options for organoid culture modes, including the following five categories: matrigel culture, polymer scaffold culture, rotary cell culture system (RCCS), air–liquid interface (ALI), and microfluidic chip technology (Figure 1).

The matrigel culture is the most classical culture mode. At 0–4 °C, the liquid matrigel and cells are mixed thoroughly and implanted into the pre-warmed well plates, then placed at 37 °C. Matrigel can be solidified to form a dome, causing the cells to be suspended in it. Finally, the relevant medium is added slowly until the dome is completely submerged. This culture mode is relatively simple and inexpensive, but the spatial relationship between the extracellular matrix and cells cannot be artificially adjusted [58,59].

Polymer scaffolds belong to the synthetic class of matrix, which can be used to artificially construct the structure of the extracellular matrix and adjust the spatial position of cells using 3D bioprinting technology. Therefore, polymer scaffolds offer greater moldability for the extracellular matrix, but are relatively costly [57].

The RCCS was originally developed by NASA. It uses the microgravity effect to create a liquid suspension in the horizontal direction so that the cells do not collide with the tube walls. The cells can maintain maximum activity throughout the culture process because of no air bubbles and no propellers [60]. It was found that RCCS was effective in improving the proliferative capacity and extending the time window for the organoids’ activity [61].

ALI refers to a culture process in which one part of the culture is exposed to air and the other part is in contact with the culture medium. This type of culture mode can provide a high oxygen concentration to the culture system and is suitable for the culture of respiratory, brain and skin organoids [62,63].

As the latest advances in bioengineering technology, microfluidic chips are integrated and assembled from micro-pumps, micro-valves and micro-channels to enable precise regulation of fluids [64]. Traditional 3D organoids cannot establish vascular tissues (such as blood and lymphatic vessels) and the matrigel or polymer scaffolds in organoids also limit the intervention and detection of cultures. A microfluidic chip-based organoid culture mode has been developed. Compared with traditional culture models, microfluidic chips have the following advantages: (1) inside the chip is a closed tube, which can avoid contamination during the culture; (2) as the chip is made of transparent materials, it is easy to monitor in real time; (3) it is suitable for high-throughput screening of drugs and targets; (4) the chip can precisely regulate the fluid, which can simulate the relationship between body fluids and tissues in vivo [65,66]. However, the chips are expensive and their application in medical studies needs further research.

## 4. Application of Gastric Organoids

### 4.1. Tissue Engineering

In recent years, the emergence of organoid models has led to the further development of tissue engineering. Organoid models can effectively restore the physiological and pathological characteristics of counterparts in vitro. According to the histological type of the counterparts, organoid models can be divided into ectodermal organoids, mesodermal organoids and ectodermal organoids [67,68,69]. The ectoderm is distributed in the outer layer of the embryo and can differentiate into skin and brain. The mesoderm is the source of muscles, vascular system, glands, kidneys, etc. Organoid models from the ectoderm and mesoderm have been constructed [70,71,72,73]. Gastric organoids belong to the endodermal organoids and have a more complex cell type. The major components of the gastric organoids are various gastric epithelial cells, in addition to Lg5^+^ cells and gastric endocrine cells [74,75,76]. It was found that the gastric organoids were highly similar to the source tissues in terms of genetic characteristics [18]. Therefore, gastric organoids become the best vehicle for gene editing to perform functional experiments. Lo YH et al. obtained *ARID1A*-deficient gastric organoids using CRISPR/Cas9 and demonstrated the effect of *ARID1A*-deficient in early-stage gastric cancer [27]. Nanki K et al. used gene editing technology to study the genotype–phenotype correlations in gastric organoids [17].

Currently, in addition to adult stem cells, hPSC can also be used as a source of organoid culture. It was found that gastric organoids can not only maintain expansion in vitro, but also differentiate to mature gastric epithelial cells under specific culture conditions. It can provide a platform for simulating various gastric diseases in vitro [67]. The organoids can be used as potential donors for transplantation. It was found that transplantation of mouse colonic organoids into the mouse model of radiation proctitis effectively stimulated the regeneration of epithelial cells and effectively improved colonic injury after radiotherapy [77]. Similar effects also occurred in the mouse endometrium. It is found that implantation of analogous organs in the model of intrauterine adhesion (IUA) was effective in inducing angiogenesis and promoting recovery of endometrial function [30]. The ultimate goal of organoids is to generate functional human organs for regenerative medicine. Fotios Sampaziotis et al. used a new model of cell engraftment in human livers to demonstrate that extrahepatic organoids can rapidly and effectively repair the intrahepatic bile duct epithelium [78]. Thus, organoid model technology has great potential in regenerative medicine research. However, organoid technology has not yet been applied to humans and challenges are still present. Firstly, there is a significant difference between the volume of the organoid model and the stomach, and it is still a technical challenge to culture the organoid to the same volume as the organ [79]. Secondly, current gastric organoids still use adult stem cells as a source and therefore, can only proliferate and differentiate to produce epithelial-derived cells, while lacking various mesenchymal-derived cells. However, human pluripotent stem cells (hPSCs) could help to revolutionize the culture paradigm of gastric organoids. hPSCs proliferate in vitro to produce a large number of visceral mesenchymal cells, which could be the main source of mesenchymal cells for gastric organoids. It was demonstrated that mixing mesenchymal cells in mouse gastric organoid models make the gastric organoids more similar to gastric tissue in terms of histology and function [28,80].

### 4.2. Gastrointestinal Cancer–Microorganism Interaction

Gastrointestinal organoids have provided new ideas for studying the relationship between microorganisms and hosts. hPSCs-derived intestinal organoids have been used by Finkbeiner et al. to model the process of viral infection. The organoid models are not only applicable to the study of viruses, but also to the exploration of other microorganisms [81]. Forbester et al. simulated the process of bacterial damage to epithelial cells by injecting *Salmonella enterica* into the lumen of the hPSCs-drived intestinal organoids [82]. This shows that organoid models have many natural advantages for studying microorganisms [83].

It is universally acknowledged that *Helicobacter pylori* (*H. pylori*) infection is associated with various gastric-related diseases, such as peptic ulcer and gastric cancer, which have been demonstrated to be directly related to *H. pylori* [84,85]. Studies have found that the proportion of patients with *H. pylori* infection who develop peptic ulcers and gastric cancer is approximately 10–20%. Among peptic ulcer patients, 95% of patients with duodenal ulcers and 80% of patients with gastric ulcers were positive for *H. pylori* infection [86,87]. To further investigate the mechanism of both, researchers injected *H. pylori* into the lumen of gastric organoids through microinjection [74,88], and found that bacterial colonization led to increased proliferation of Lgr5 stem cells, which was induced by the bacterial virulence factor *CagA* [89,90]. Although this result has been demonstrated in 2D cell line experiments, there are still problems due to the single cell type in the cell line and the fact that the 2D growth pattern is also very different from the in vivo cells. Only adding bacteria to the medium does not reflect the interaction between bacteria and host well, but the gastric organoid microinjection technique can address this issue very well [91]. Moreover, *H. pylori* was found to induce cell differentiation and activate both the NF-κB pathway and the chemokine IL-8, allowing for an increased inflammatory response, and urea production by the epithelial cell wall was found to be necessary for *H. pylori* colonization of the gastric mucosa in gastric organoids [92]. Gastric organoids also have been used for microenvironmental and *H. pylori* studies. Using gastric organoid models, researchers found that spermine oxidase (SMOX) can cause inflammation, DNA damage, and thus promote *H. pylori*-induced carcinogenesis [21].

Co-culture systems of immune cells and gastric organoids may reveal more novel pathological mechanisms of *H. pylori*. Using a co-culture system of gastric organoids and immune cells, researchers found that PD-L1 production was significantly increased in *H. pylori*-infected groups, and this process was associated with Sonic Hedgehog (Shh) signaling pathway [93]. Gastric organoids can restore the immune response of immune cells in response to inflammatory stimulation. It was found that *H. pylori* infection increased chemokines in the gastric organoids, leading to the migration of DCs to specific areas and phagocytosis of *H. pylori*, a process very similar to the in vivo mechanism [94].

Therefore, gastric organoid models are useful not only for studying the pathogenesis of *H. pylori*, but also for understanding the role of microorganisms in diseases such as cancer and inflammatory bowel disease (IBD) by helping to characterize the host’s response to various microbial colonization.

### 4.3. Drug Testing and Development

Gastric organoid is a reliable model for testing drugs. Anti-tumor drugs for gastric cancer can be classified into chemotherapeutic drugs and targeted drugs. Chemotherapy drugs are the most basic and important part of treatment, such as cisplatin, 5-fluorouracil (5-FU), irinotecan, and oxaliplatin. It has been found that the IC_50_ of various chemotherapeutic drugs tested in PDO is consistent with the results obtained in many clinical pharmacology studies [15,30]; that is, the same drug can achieve consistent pharmacological effects in organoid models and in vivo. Similarly, various targeted drugs show the same effects in PDO [32]. The reason for this may be that gastric organoids can maintain a large proportion of the genetic characteristics of cancer tissues, such as various mutated genes. Researchers performed whole exome sequencing (WES) analysis on 12 groups of gastric organoid models and tumor tissues and found highly similar various mutated genes between them. In particular, within the mutated genes, the proportion of different base mutations was significantly similar between the two groups [30]. Therefore, PDO, as a biological model, can be trusted for drug testing in vitro. In the recent five years, many research studies have used gastric organoids for drug testing (Table 2).

Gastric organoids are more efficient in drug testing and development, such as detecting drug toxicity, developing new drugs and high-throughput screening [32]. Traditional screening of anti-tumor drugs is a complex process, whereas drug screening using a PDO model can address some of the drawbacks of traditional methods. The traditional method is often based on the construction of animal models, so the screening process is more time-consuming and expensive, and the experimental cycle is longer. PDO models are derived from the patient, and more beneficial for individualized treatment. The 3D cell culture mode has a higher throughput, allowing for the screening of multiple drugs at the same time [95]. However, most of the current studies are still validating the reliability of PDO, in which new drug development is less reported.

## 5. Limitations of Organoid Models

The organoid models are one of the most novel biological models in the last years, but they also have some limitations. Firstly, the human internal environment is more complex and difficult to restore entirely in vitro. In addition to nutritional support, the growth of many cells and tissues is often regulated by changes in the concentration of growth factors [96]. Secondly, obtaining pure cancer organoids is currently still difficult because tumor organoid may be mixed with overgrown healthy epithelial-derived organoids [97]. Researchers have now begun to focus on tumor organoid screening, which may be one of the next research directions [98]. Thirdly, current organoid models are unable to construct a well-developed vascular system, resulting in organoid models that cannot be cultured to a larger size in vitro. The development of engineering vascularization may overcome this problem [99]. Finally, organoid models are highly influenced by the culture environment and have poor reproducibility [100].

## 6. Concluding Remarks and Future Perspective

In recent years, various organoid models have already been constructed. Current studies on organoids not only focus on the optimization of construction methods, but also on the biological applications and medical translation of organoids [96,101]. Organoids have unique advantages over classical biological models in vitro. Cell lines are the most commonly used biological models, obtained by artificial in vitro culture. However, due to multiple passages, cell function and growth mode are different from in vivo cells, and there is a lack of tumor heterogeneity. In contrast, organoid models can better maintain tumor heterogeneity in vivo. Patient-derived tumor xenografts (PDX) are models constructed by implanting primary cells from patient-derived tumor tissue into immunodeficient mice. PDO models extract stem cells from the tissue for in vitro expansion. Both are biological models using patient-derived tissues, but there are still some differences between them: (1) tumor microenvironment: PDX models use animals as ‘culture media’, while POD models use culture media made to simulate the tumor microenvironment; (2) PDX models are time-consuming and expensive, but the culture cycle of POD is relatively short. Therefore, the POD models are more suitable for drug screening and tumor target research because of less expensive culture costs and a larger number of cultured cells [96,102].

Three-dimensional bioprinting technology could provide new ideas for the construction of organoid models. The core of in vitro 3D culture technology lies in the restored microenvironment. The tumor microenvironment (TME) is more complex, and in addition to various cells, stromal components such as fibroblasts are the key to build the TME, so how to restore the structure of the matrix in vitro is another challenge for organoid models. Three-dimensional bioprinting technology can utilize various biological materials to complete the construction of a 3D skeleton with human design [103]. Therefore, 3D bioprinting technology can make it possible to artificially control the 3D structure of an organoid. The concept of ‘next-generation’ organoids has been proposed, in which certain types of cells require extra signaling factors in addition to basic nutrients and cytokines during the proliferation and differentiation. Therefore, researchers have proposed the use of bioelectrical signals in the microenvironment to build brain organoids and retinal organoids [104]. Opportunities and challenges remain in the studies of gastric organoids. Researchers are currently bringing forward the use of tumor organoids to establish individualized in vitro model systems to investigate interactions between T cells and tumor cells for immunotherapy of gastric cancer [105]. Nevertheless, there are still limitations of the gastric organoids, such as the possibility of losing the ability to differentiate into functional cells after multiple passages [106]. In addition to this, matrix gels often stem from other species that are not completely removed during subsequent studies, which may affect the results of the studies [97].

In summary, gastric organoid models have seen advances in stem cell culture techniques, extracellular matrix development, and optimization of 3D culture methods. Meanwhile, gastric organoids are also widely used as biological models in medical research. The organoid model still has many limitations and needs further study. However, it is undeniable that gastric organoids provide a novel idea for the study of gastric-related diseases, offer a new way for individualized treatment, and promise to be an important research tool.

## Figures and Tables

**Figure 1 biomolecules-13-00875-f001:**
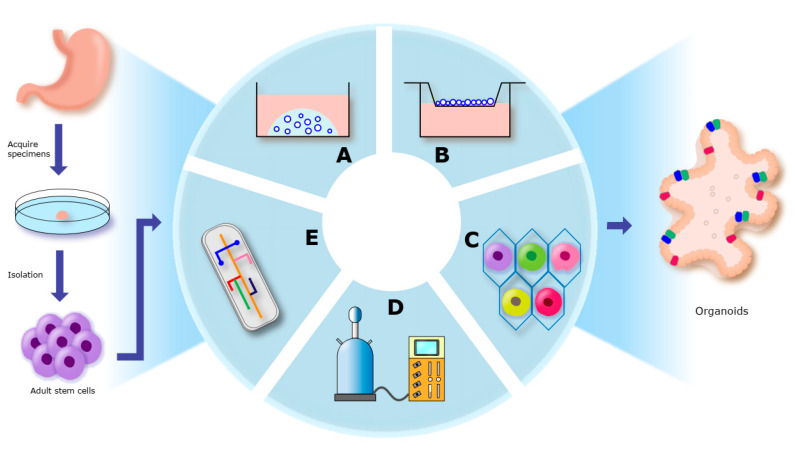
Culture modes of PDO in gastric cancer. (**A**): Matrigel culture. (**B**): Air-liquid interface. (**C**): Polymer scaffold culture. (**D**): Rotary cell culture system. (**E**): Microfluidic chip technology.

**Table 1 biomolecules-13-00875-t001:** Gastric organoids from different stem cells.

	iPSC-Derived	ASC-Derived
Cell types	Epithelial and mesenchymal cells	Epithelial cells only
Proliferative capacity in vitro	Multiple passages	Unlimited multiplication
Modelling time	4–8 weeks	1–2 weeks
Histomorphological features	Resembling embryonic tissue	Resembling adult tissue

**Table 2 biomolecules-13-00875-t002:** Summary of studies on gastric organoids in the last five years.

Year	Authors	Tissue Acquisition	Stem Cells	Methods of Isolating Glands	Applications
2018	Gao M et al. [15]	Endoscope Surgery	ASC	Physical pressure	Next-generation sequencing Drug testing
2018	Vlachogiannis G et al. [16]	Biopsy	ASC	TrypLE	Whole genome sequencing Drug testing Targeted therapy testing Biobank establishment
2018	Nanki K et al. [17]	Surgery Endoscope Ascites puncture	ASC	Libase TH TrypLE	CRISPR/Cas9 Transcriptomic analysis Xenotransplantation of organoids Biobank establishment
2019	Steele NG et al. [18]	Surgery	ASC	Collagenase Hyaluronidase	RNA sequencing Drug testing Targeted therapy testing Organoids tumorigenicity analysis
2019	Li J et al. [19]	Ascites puncture	ASC	-	Malignant ascites-derived organoid Drug testing
2019	Seidlitz T et al. [20]	Surgery	ASC	Collagenase XI Dispase II	Whole genome sequencing RNA sequencing Drug testing Targeted therapy testing Biobank establishment
2020	Sierra JC et al. [21]	Surgery	ASC	Collagenase IV	*H. pylori*
2021	Giobbe GG. et al. [22]	Biopsy	ASC	Physical pressure	SARS-CoV-2 infection
2021	Gobert AP et al. [23]	Endoscope	ASC	-	*H. pylori*
2021	Koh V et al. [24]	Surgery	ASC	Collagenase	Immunoregulation in EMT
2021	Togasaki K et al. [25]	Surgery Ascites puncture	ASC	Libase TH	Whole exome sequence RNA sequence Xenotransplantation of organoids
2021	Chakrabarti J et al. [26]	Biopsy	ASC	Collagenase I Hyaluronidase IV-S	Organoid/immune cells co-culture model Targeted therapy testing
2021	Lo YH et al. [27]	Surgery	ASC	Collagenase I	CRISPR/Cas9
2022	Eicher AK et al. [28]	-	hPSC	-	Organoid assembly approach Tissue engineering
2022	Miao X et al. [29]	Surgery	ASC	Collagenase II Dispase II	Drug testing
2022	Li G et al. [30]	Surgery	ASC	Collagenase	Organoids tumorigenicity analysis Drug testing
2023	Yoon C et al. [31]	Endoscope	ASC	Collagenase III	Drug testing
2023	Zhang H et al. [32]	Surgery	ASC	Gentle MACS™ Dissociator tumor dissociation kit	Transcriptome sequencing Drug testing

## Data Availability

Not applicable.

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
