# Peer review of "Organoids: Construction and Application in Gastric Cancer"

_biomolecules, 2023, doi:10.3390/biom13050875_

Round 1
Reviewer 1 Report
Huo C et al have provided a very compressive review on organoids; revising not only the gastric cancer organoids, but all of them, with technical and historical insight. I really enjoyed the approach. Few more figures would help the not expert reader, for example at the end of the paragraph 3.1 a table or a figure could help to follow the different source of cells to produce organoids. There are e few repetition such as in the beginning of the 3.1 paragraph the word "derived" is repeated 3 times.
Few sentences are not clear to me, such as in the paragraph 3.2.2 by the end there is the sentence "It can be constructed artificially on extracellular matrix..." What can be constructed?
Overall I think this review is informative and original, I strongly encourage its publication.
Author Response
Thank you for your careful review. We really appreciate your efforts in reviewing our manuscript. Your careful review has helped to make our study clearer and more comprehensive.
Comment 1: Few sentences are not clear to me, such as in the paragraph 3.2.2 by the end there is the sentence "It can be constructed artificially on extracellular matrix..." What can be constructed?
Response 1: Thank you very much for your important reminder. 'Composite' matrix material is a new technology that combines natural matrix and synthetic matrix. By using fibroblasts implanted in synthetic scaffolds, we will obtain the new material containing two matrices at the same time. We have revised this part of the article as follows:
Currently, researchers have developed 'composite' matrix material, which owns two types of extracellular matrix. This material is obtained by implanting fibroblasts inside the synthetic scaffolds, using the fibroblasts to produce natural matrix, and removing the fibroblasts afterwards. 'Composite' matrix material is more conducive to the generation of organoids and has a much shorter modeling time compared to one matrix alone [39,40].
- Nayak, B.; Balachander, G.M.; Manjunath, S.; Rangarajan, A.; Chatterjee, K. Tissue mimetic 3D scaffold for breast tumor-derived organoid culture toward personalized chemotherapy. Colloids and surfaces. B, Biointerfaces 2019, 180, 334-343, doi:10.1016/j.colsurfb.2019.04.056.
- Balachander, G.M.; Balaji, S.A.; Rangarajan, A.; Chatterjee, K. Correction to "Enhanced Metastatic Potential in a 3D Tissue Scaffold toward a Comprehensive in Vitro Model for Breast Cancer Metastasis". ACS applied materials & interfaces 2021, 13, 41361, doi:10.1021/acsami.1c14650.
Comment 2: There are e few repetitions such as in the beginning of the 3.1 paragraph the word "derived" is repeated 3 times.
Response 2: Thank you for your precious advice. We have corrected this error in the initial location as follows:
The construction of organoid models is based on stem cell culture techniques. Gastric cancer organoids can be derived fromadult stem cells (ASC), induced pluripotent stem cells (iPSC), and embryonic stem cells (ESC).
Attached is the manuscript of our revised model.

Reviewer 2 Report
The aforementioned article deals with a very interesting issuue regarding the role of organoids in gastric cancer. The stdy is well established with an adequate approach to the organoids in general but the authors do not refer exensively to the role of organoids in relation to gastric cancer and how can affect the treatment options.
Author Response
Thank you for your careful review. We really appreciate your efforts in reviewing our manuscript. Your careful review has helped to make our study clearer and more comprehensive. Attached is the manuscript of our revised model.
Comment 1: Authors do not refer exensively to the role of organoids in relation to gastric cancer and how can affect the treatment options.
Response 1: Thank you for your precious advice. We have noticed the deficiencies of our paper and modified the section "Application of gastric organoids"as follows:
4.1 We added some contents at the end of paragraph: It was found that the gastric organoids were highly similar to the source tissues in terms of genetic characteristics [65]. Therefore, gastric organoids become the best vehicle for gene editing to perform functional experiments. Lo YH et al. obtained ARID1A-deficient gastric organoids using CRISPR/Cas9, and demonstrated the effect of ARID1A-deficient in early-stage gastric cancer [66]. Nanki K et al. used gene editing technology to study the genotype-phenotype correlations in gastric organoids [67].
4.2 The subtitle has already changed into “Gastrointestinal cancer-microorganism interaction”. And we added some contents at the end of paragraph: Gastric organoids also be used for microenvironmental and H. pylori studies. Using gastric organoid models, researchers found that spermine oxidase (SMOX) can cause inflammation, DNA damage, and thus promote H. pylori-induced carcinogenesis [86].
4.3 We rewrote this section and added Table 2 about the important articles related to gastric organoids in the last five years.
Gastric organoid is a reliable model for testing drugs. Anti-tumor drugs for gastric cancer can be classified into chemotherapeutic drugs and targeted drugs. Chemotherapy drugs are the most basic and important part of treatment, such as cisplatin, 5-fluorouracil (5-FU), irinotecan, and oxaliplatin. It has been found that the IC50 of various chemotherapeutic drugs tested in PDO is consistent with the results obtained in many clinical pharmacology studies [15,69], that is, the same drug can achieve consistent pharmacological effects in organoid models and in vivo. Similarly, various targeted drugs show the same effects in PDO [90]. The reason for this may be because gastric organoids can maintain a large proportion of the genetic characteristics of cancer tissues, such as various mutated genes. Researchers performed whole-exome sequencing (WES) analysis on 12 groups of gastric organoid models and tumor tissues, and found highly similar various mutated genes between them. In particular, within the mutated genes, the proportion of different base mutations was significantly similar between the two groups [69]. Therefore, PDO, as a biological model, can be trust for drug testing in vitro. In the recent five years, many researches used gastric organoids for drug testing (Table 2).
Gastric organoids are more efficient in drug testing and development such as detecting drug toxicity, developing new drugs and high-throughput screening [84]. Traditional screening of anti-tumor drugs is a complex process, whereas drug screening using PDO model can address some of the drawbacks of traditional methods. The traditional method is often based on the construction of animal models, so the screening process is more time-consuming and expensive, and the experimental cycle is longer. PDO models are derived from the patient, and more beneficial for individualized treatment. The 3D cell culture mode has a higher throughput, allowing for the screening of multiple drugs at the same time [91]. However, most of the current studies are still validating the reliability of PDO, in which new drug development is less reported.
65. Steele, N.G.; Chakrabarti, J.; Wang, J.; Biesiada, J.; Holokai, L.; Chang, J.; Nowacki, L.M.; Hawkins, J.; Mahe, M.; Sundaram, N.; et al. An Organoid-Based Preclinical Model of Human Gastric Cancer. Cellular and molecular gastroenterology and hepatology 2019, 7, 161-184, doi:10.1016/j.jcmgh.2018.09.008.
66. Lo, Y.H.; Kolahi, K.S.; Du, Y.; Chang, C.Y.; Krokhotin, A.; Nair, A.; Sobba, W.D.; Karlsson, K.; Jones, S.J.; Longacre, T.A.; et al. A CRISPR/Cas9-Engineered ARID1A-Deficient Human Gastric Cancer Organoid Model Reveals Essential and Nonessential Modes of Oncogenic Transformation. Cancer discovery 2021, 11, 1562-1581, doi:10.1158/2159-8290.Cd-20-1109.
67. Nanki, K.; Toshimitsu, K.; Takano, A.; Fujii, M.; Shimokawa, M.; Ohta, Y.; Matano, M.; Seino, T.; Nishikori, S.; Ishikawa, K.; et al. Divergent Routes toward Wnt and R-spondin Niche Independency during Human Gastric Carcinogenesis. Cell 2018, 174, 856-869.e817, doi:10.1016/j.cell.2018.07.027.
86. Sierra, J.C.; Piazuelo, M.B.; Luis, P.B.; Barry, D.P.; Allaman, M.M.; Asim, M.; Sebrell, T.A.; Finley, J.L.; Rose, K.L.; Hill, S.; et al. Spermine oxidase mediates Helicobacter pylori-induced gastric inflammation, DNA damage, and carcinogenic signaling. Oncogene 2020, 39, 4465-4474, doi:10.1038/s41388-020-1304-6.
15. Gao, M.; Lin, M.; Rao, M.; Thompson, H.; Hirai, K.; Choi, M.; Georgakis, G.V.; Sasson, A.R.; Bucobo, J.C.; Tzimas, D.; et al. Development of Patient-Derived Gastric Cancer Organoids from Endoscopic Biopsies and Surgical Tissues. Annals of surgical oncology 2018, 25, 2767-2775, doi:10.1245/s10434-018-6662-8.
69. Li, G.; Ma, S.; Wu, Q.; Kong, D.; Yang, Z.; Gu, Z.; Feng, L.; Zhang, K.; Cheng, S.; Tian, Y.; et al. Establishment of gastric signet ring cell carcinoma organoid for the therapeutic drug testing. Cell death discovery 2022, 8, 6, doi:10.1038/s41420-021-00803-7.
90. Zhang, H.; Qin, Y.; Jia, M.; Li, L.; Zhang, W.; Li, L.; Zhang, Z.; Liu, Y. A gastric cancer patient-derived three-dimensional cell spheroid culture model. American journal of cancer research 2023, 13, 964-975.
91. Seidlitz, T.; Koo, B.K.; Stange, D.E. Gastric organoids-an in vitro model system for the study of gastric development and road to personalized medicine. Cell death and differentiation 2021, 28, 68-83, doi:10.1038/s41418-020-00662-2.
Table 2. Summary of papers on gastric organoids in the last five years
|
Authors |
Tissue acquisition |
Stem cells |
Methods of isolating glands |
Applications |
|
Gao M et al. [15] |
Endoscope Surgery |
ASC |
Physical pressure |
Next-generation sequencing Drugs testing |
|
Li J et al.[102] |
Ascites puncture |
ASC |
- |
Malignant ascites-derived organoid Drugs testing |
|
Seidlitz T et al.[103] |
Surgery |
ASC |
Collagenase XI Dispase II |
Whole genome sequencing RNA sequencing Drugs testing Targeted therapy testing Biobank establishment |
|
Giobbe GG. et al.[104] |
Biopsy |
ASC |
Physical pressure |
SARS-CoV-2 infection |
|
Vlachogiannis G et al.[105] |
Biopsy |
ASC |
TrypLE |
Whole genome sequencing Drugs testing Targeted therapy testing Biobank establishment |
|
Nanki K et al.[67] |
Surgery Endoscope Ascites puncture |
ASC |
Libase TH TrypLE |
CRISPR/Cas9 Transcriptomic analysis Xenotransplantation of Organoids Biobank establishment |
|
Steele NG et al.[65] |
Surgery |
ASC |
Collagenase Hyaluronidase |
RNA sequencing Drugs testing Targeted therapy testing Organoids tumorigenicity analysis |
|
Gobert AP et al.[106] |
Endoscope |
ASC |
- |
H. pylori |
|
Eicher AK et al.[73] |
- |
hPSC |
- |
Organoid assembly approach Tissue engineering |
|
Sierra JC et al. [86] |
Surgery |
ASC |
Collagenase IV |
H. pylori |
|
Koh V et al.[107] |
Surgery |
ASC |
Collagenase |
Immunoregulation in EMT |
|
Togasaki K et al. [108] |
Surgery Ascites puncture |
ASC |
Libase TH |
Whole-exome Sequence RNA Sequence Xenotransplantation of Organoids |
|
Chakrabarti J et al. [109] |
Biopsy |
ASC |
Collagenase I Hyaluronidase IV-S |
Organoid/immune cells co-culture model Targeted therapy testing |
|
Lo YH et al. [66] |
Surgery |
ASC |
Collagenase I |
CRISPR/Cas9 |
|
Miao X et al. [110] |
Surgery |
ASC |
Collagenase II Dispase II |
Drugs testing |
|
Li G et al. [69] |
Surgery |
ASC |
Collagenase |
Organoids tumorigenicity analysis Drugs testing |
|
Yoon C et al. [111] |
Endoscope |
ASC |
Collagenase III |
Drugs testing |
|
Zhang H et al.[90] |
Surgery |
ASC |
Gentle MACS™ Dissociator tumor dissociation kit |
Transcriptome sequencing Drugs testing |
15. Gao, M.; Lin, M.; Rao, M.; Thompson, H.; Hirai, K.; Choi, M.; Georgakis, G.V.; Sasson, A.R.; Bucobo, J.C.; Tzimas, D.; et al. Development of Patient-Derived Gastric Cancer Organoids from Endoscopic Biopsies and Surgical Tissues. Annals of surgical oncology 2018, 25, 2767-2775, doi:10.1245/s10434-018-6662-8.
102. Li, J.; Xu, H.; Zhang, L.; Song, L.; Feng, D.; Peng, X.; Wu, M.; Zou, Y.; Wang, B.; Zhan, L.; et al. Malignant ascites-derived organoid (MADO) cultures for gastric cancer in vitro modelling and drug screening. Journal of cancer research and clinical oncology 2019, 145, 2637-2647, doi:10.1007/s00432-019-03004-z.
103. Seidlitz, T.; Merker, S.R.; Rothe, A.; Zakrzewski, F.; von Neubeck, C.; Grützmann, K.; Sommer, U.; Schweitzer, C.; Schölch, S.; Uhlemann, H.; et al. Human gastric cancer modelling using organoids. Gut 2019, 68, 207-217, doi:10.1136/gutjnl-2017-314549.
104. Giobbe, G.G.; Bonfante, F.; Jones, B.C.; Gagliano, O.; Luni, C.; Zambaiti, E.; Perin, S.; Laterza, C.; Busslinger, G.; Stuart, H.; et al. SARS-CoV-2 infection and replication in human gastric organoids. Nature communications 2021, 12, 6610, doi:10.1038/s41467-021-26762-2.
105. Vlachogiannis, G.; Hedayat, S.; Vatsiou, A.; Jamin, Y.; Fernández-Mateos, J.; Khan, K.; Lampis, A.; Eason, K.; Huntingford, I.; Burke, R.; et al. Patient-derived organoids model treatment response of metastatic gastrointestinal cancers. Science (New York, N.Y.) 2018, 359, 920-926, doi:10.1126/science.aao2774.
67. Nanki, K.; Toshimitsu, K.; Takano, A.; Fujii, M.; Shimokawa, M.; Ohta, Y.; Matano, M.; Seino, T.; Nishikori, S.; Ishikawa, K.; et al. Divergent Routes toward Wnt and R-spondin Niche Independency during Human Gastric Carcinogenesis. Cell 2018, 174, 856-869.e817, doi:10.1016/j.cell.2018.07.027.
65. Steele, N.G.; Chakrabarti, J.; Wang, J.; Biesiada, J.; Holokai, L.; Chang, J.; Nowacki, L.M.; Hawkins, J.; Mahe, M.; Sundaram, N.; et al. An Organoid-Based Preclinical Model of Human Gastric Cancer. Cellular and molecular gastroenterology and hepatology 2019, 7, 161-184, doi:10.1016/j.jcmgh.2018.09.008.
106. Gobert, A.P.; Boutaud, O.; Asim, M.; Zagol-Ikapitte, I.A.; Delgado, A.G.; Latour, Y.L.; Finley, J.L.; Singh, K.; Verriere, T.G.; Allaman, M.M.; et al. Dicarbonyl Electrophiles Mediate Inflammation-Induced Gastrointestinal Carcinogenesis. Gastroenterology 2021, 160, 1256-1268.e1259, doi:10.1053/j.gastro.2020.11.006.
73. Eicher, A.K.; Kechele, D.O.; Sundaram, N.; Berns, H.M.; Poling, H.M.; Haines, L.E.; Sanchez, J.G.; Kishimoto, K.; Krishnamurthy, M.; Han, L.; et al. Functional human gastrointestinal organoids can be engineered from three primary germ layers derived separately from pluripotent stem cells. Cell stem cell 2022, 29, 36-51.e36, doi:10.1016/j.stem.2021.10.010.
86. Sierra, J.C.; Piazuelo, M.B.; Luis, P.B.; Barry, D.P.; Allaman, M.M.; Asim, M.; Sebrell, T.A.; Finley, J.L.; Rose, K.L.; Hill, S.; et al. Spermine oxidase mediates Helicobacter pylori-induced gastric inflammation, DNA damage, and carcinogenic signaling. Oncogene 2020, 39, 4465-4474, doi:10.1038/s41388-020-1304-6.
107. Koh, V.; Chakrabarti, J.; Torvund, M.; Steele, N.; Hawkins, J.A.; Ito, Y.; Wang, J.; Helmrath, M.A.; Merchant, J.L.; Ahmed, S.A.; et al. Hedgehog transcriptional effector GLI mediates mTOR-Induced PD-L1 expression in gastric cancer organoids. Cancer letters 2021, 518, 59-71, doi:10.1016/j.canlet.2021.06.007.
108. Togasaki, K.; Sugimoto, S.; Ohta, Y.; Nanki, K.; Matano, M.; Takahashi, S.; Fujii, M.; Kanai, T.; Sato, T. Wnt Signaling Shapes the Histologic Variation in Diffuse Gastric Cancer. Gastroenterology 2021, 160, 823-830, doi:10.1053/j.gastro.2020.10.047.
109. Chakrabarti, J.; Koh, V.; So, J.B.Y.; Yong, W.P.; Zavros, Y. A Preclinical Human-Derived Autologous Gastric Cancer Organoid/Immune Cell Co-Culture Model to Predict the Efficacy of Targeted Therapies. Journal of visualized experiments : JoVE 2021, doi:10.3791/61443.
109. Chakrabarti, J.; Koh, V.; So, J.B.Y.; Yong, W.P.; Zavros, Y. A Preclinical Human-Derived Autologous Gastric Cancer Organoid/Immune Cell Co-Culture Model to Predict the Efficacy of Targeted Therapies. Journal of visualized experiments : JoVE 2021, doi:10.3791/61443.
66. Lo, Y.H.; Kolahi, K.S.; Du, Y.; Chang, C.Y.; Krokhotin, A.; Nair, A.; Sobba, W.D.; Karlsson, K.; Jones, S.J.; Longacre, T.A.; et al. A CRISPR/Cas9-Engineered ARID1A-Deficient Human Gastric Cancer Organoid Model Reveals Essential and Nonessential Modes of Oncogenic Transformation. Cancer discovery 2021, 11, 1562-1581, doi:10.1158/2159-8290.Cd-20-1109.
110. Miao, X.; Wang, C.; Chai, C.; Tang, H.; Hu, J.; Zhao, Z.; Luo, W.; Zhang, H.; Zhu, K.; Zhou, W.; et al. Establishment of gastric cancer organoid and its application in individualized therapy. Oncology letters 2022, 24, 447, doi:10.3892/ol.2022.13567.
69. Li, G.; Ma, S.; Wu, Q.; Kong, D.; Yang, Z.; Gu, Z.; Feng, L.; Zhang, K.; Cheng, S.; Tian, Y.; et al. Establishment of gastric signet ring cell carcinoma organoid for the therapeutic drug testing. Cell death discovery 2022, 8, 6, doi:10.1038/s41420-021-00803-7.
111. Yoon, C.; Lu, J.; Kim, B.J.; Cho, S.J.; Kim, J.H.; Moy, R.H.; Ryeom, S.W.; Yoon, S.S. Patient-Derived Organoids from Locally Advanced Gastric Adenocarcinomas Can Predict Resistance to Neoadjuvant Chemotherapy. Journal of gastrointestinal surgery : official journal of the Society for Surgery of the Alimentary Tract 2023, 27, 666-676, doi:10.1007/s11605-022-05568-7.

Reviewer 3 Report
In the review article “Organoids: Construction and Application in Gastric Cancer” reviewed the organoids
·The article discusses the use of organoids in gastric cancer research. Although there are many previous reviews that have addressed a similar issue in other cancers, the current article discusses the use and limitation of organoids specific to gastric cancer. Although the discussion about the use of organoids in gastrointestinal cancer and microbiome interaction, overall, the writing is more general. More specific discussion about what specific kind of cancer-microenvironment interaction can be studied using organoids will be interesting to discuss. Similarly, what species of microbiome-cancer interaction can be studied using organoids?
· The references are appropriate. However, authors should cite some important previous reviews (https://doi.org/10.1080/17460441.2022.1991306, doi: 10.1016/j.xpro.2021.101079 ) on the use of organoids in drug screening.
· Subtitle 4.2 Gastrointestinal microorganisms seem unclear. “Study the role of Gastrointestinal microorganisms in GI” or “Gastrointestinal cancer-microorganism interaction” may be better.
· Author should comment on the variability of organoid models while using them in experiments.
Author Response
Thank you for your careful review. We really appreciate your efforts in reviewing our manuscript. Your careful review has helped to make our study clearer and more comprehensive. Attached is the manuscript of our revised model.
Comment 1: More specific discussion about what specific kind of cancer-microenvironment interaction can be studied using organoids will be interesting to discuss. Similarly, what species of microbiome-cancer interaction can be studied using organoids?
Response 1: Thank you very much for your important reminder. We thought the subtitle of 4.2 “Gastrointrstinal microorganisms” is inaccurate, so we revised it to “Gastrointestinal cancer-microorganism interaction”. In this section, we mainly show the studies on some pathogenic bacteria using organoids, such as Salmonella enterica in paragraph 1, especially H. pylori in paragraph 2. And we added some contents at the end of paragraph 2: Gastric organoids also be used for microenvironmental and H. pylori studies. Using gastric organoid models, researchers found that spermine oxidase (SMOX) can cause inflammation, DNA damage, and thus promote H. pylori-induced carcinogenesis[86].
Comment 2: Authors should cite some important previous reviews on the use of organoids in drug screening.
Response 2: Thank you for your precious advice. We rewrote the section ‘4.3 Drug testing and development’. First, we present the reliability of the gastric organoid. As a still novel biological model, it is essential to verify the reliability. Next, we review the advantages of gastric organoid for drug detection. We have updated the most recent papers and focused on the application of gastric organoids. Also, we have summarized the important articles related to gastric organoids in the last five years and added Table 2 as follows:
4.3 Drug testing and development
Gastric organoid is a reliable model for testing drugs. Anti-tumor drugs for gastric cancer can be classified into chemotherapeutic drugs and targeted drugs. Chemotherapy drugs are the most basic and important part of treatment, such as cisplatin, 5-fluorouracil (5-FU), irinotecan, and oxaliplatin. It has been found that the IC50 of various chemotherapeutic drugs tested in PDO is consistent with the results obtained in many clinical pharmacology studies[15,69], that is, the same drug can achieve consistent pharmacological effects in organoid models and in vivo. Similarly, various targeted drugs show the same effects in PDO[90]. The reason for this may be because gastric organoids can maintain a large proportion of the genetic characteristics of cancer tissues, such as various mutated genes. Researchers performed whole-exome sequencing (WES) analysis on 12 groups of gastric organoid models and tumor tissues, and found highly similar various mutated genes between them. In particular, within the mutated genes, the proportion of different base mutations was significantly similar between the two groups[69]. Therefore, PDO, as a biological model, can be trust for drug testing in vitro. In the recent five years, many researches used gastric organoids for drug testing (Table 2).
Gastric organoids are more efficient in drug testing and development such as detecting drug toxicity, developing new drugs and high-throughput screening[84]. Traditional screening of anti-tumor drugs is a complex process, whereas drug screening using PDO model can address some of the drawbacks of traditional methods. The traditional method is often based on the construction of animal models, so the screening process is more time-consuming and expensive, and the experimental cycle is longer. PDO models are derived from the patient, and more beneficial for individualized treatment. The 3D cell culture mode has a higher throughput, allowing for the screening of multiple drugs at the same time[91]. However, most of the current studies are still validating the reliability of PDO, in which new drug development is less reported.
Comment 3: Subtitle 4.2 Gastrointestinal microorganisms seem unclear. “Study the role of Gastrointestinal microorganisms in GI” or “Gastrointestinal cancer-microorganism interaction” may be better.
Response 3: Thanks for your kind suggestions. Following your advice, the subtitle has been revised and marked in its original place in the manuscript.
Comment 4: Author should comment on the variability of organoid models while using them in experiments.
Response 4: Thank for your comments. We re-read the papers of the last 5 years and found some points of difference between many of the studies in gland isolation methods and tissue acquisition. We have modified the corresponding contents of the manuscript and listed these ‘other possibilities’ in Table 2, which we hope will be helpful for future readers.
3.1 Cells in organoids culture
As adult stem cells are present in the gastric glands, surgical and biopsy-obtained tissue blocks often require isolation of glands. The methods used to isolate the gland include physical pressure and enzymatic reaction. Glands in the sample can be isolated by gently pressuring the tissue using a slide. This method can better maintain the activity of the cells, but is less efficient[15]. Enzymatic reaction is the most common method of isolation, using various enzymes to break the connections between cells. This method is dependent on the type and activity of enzymes. Currently, there are different methods and enzymes used to isolate glands in many studies of gastric PDO. We summarized the studies on this issue within the last 5 years (Table 2).
Table 2. Summary of papers on gastric organoids in the last five years
|
Authors |
Tissue acquisition |
Stem cells |
Methods of isolating glands |
Applications |
|
Gao M et al. [15] |
Endoscope Surgery |
ASC |
Physical pressure |
Next-generation sequencing Drugs testing |
|
Li J et al.[102] |
Ascites puncture |
ASC |
- |
Malignant ascites-derived organoid Drugs testing |
|
Seidlitz T et al.[103] |
Surgery |
ASC |
Collagenase XI Dispase II |
Whole genome sequencing RNA sequencing Drugs testing Targeted therapy testing Biobank establishment |
|
Giobbe GG. et al.[104] |
Biopsy |
ASC |
Physical pressure |
SARS-CoV-2 infection |
|
Vlachogiannis G et al.[105] |
Biopsy |
ASC |
TrypLE |
Whole genome sequencing Drugs testing Targeted therapy testing Biobank establishment |
|
Nanki K et al.[67] |
Surgery Endoscope Ascites puncture |
ASC |
Libase TH TrypLE |
CRISPR/Cas9 Transcriptomic analysis Xenotransplantation of Organoids Biobank establishment |
|
Steele NG et al.[65] |
Surgery |
ASC |
Collagenase Hyaluronidase |
RNA sequencing Drugs testing Targeted therapy testing Organoids tumorigenicity analysis |
|
Gobert AP et al.[106] |
Endoscope |
ASC |
- |
H. pylori |
|
Eicher AK et al.[73] |
- |
hPSC |
- |
Organoid assembly approach Tissue engineering |
|
Sierra JC et al.[86] |
Surgery |
ASC |
Collagenase IV |
H. pylori |
|
Koh V et al.[107] |
Surgery |
ASC |
Collagenase |
Immunoregulation in EMT |
|
Togasaki K et al. [108] |
Surgery Ascites puncture |
ASC |
Libase TH |
Whole-exome Sequence RNA Sequence Xenotransplantation of Organoids |
|
Chakrabarti J et al. [109] |
Biopsy |
ASC |
Collagenase I Hyaluronidase IV-S |
Organoid/immune cells co-culture model Targeted therapy testing |
|
Lo YH et al.[66] |
Surgery |
ASC |
Collagenase I |
CRISPR/Cas9 |
|
Miao X et al.[110] |
Surgery |
ASC |
Collagenase II Dispase II |
Drugs testing |
|
Li G et al.[69] |
Surgery |
ASC |
Collagenase |
Organoids tumorigenicity analysis Drugs testing |
|
Yoon C et al.[111] |
Endoscope |
ASC |
Collagenase III |
Drugs testing |
|
Zhang H et al.[90] |
Surgery |
ASC |
Gentle MACS™ Dissociator tumor dissociation kit |
Transcriptome sequencing Drugs testing |
15. Gao, M.; Lin, M.; Rao, M.; Thompson, H.; Hirai, K.; Choi, M.; Georgakis, G.V.; Sasson, A.R.; Bucobo, J.C.; Tzimas, D.; et al. Development of Patient-Derived Gastric Cancer Organoids from Endoscopic Biopsies and Surgical Tissues. Annals of surgical oncology 2018, 25, 2767-2775, doi:10.1245/s10434-018-6662-8.
102. Li, J.; Xu, H.; Zhang, L.; Song, L.; Feng, D.; Peng, X.; Wu, M.; Zou, Y.; Wang, B.; Zhan, L.; et al. Malignant ascites-derived organoid (MADO) cultures for gastric cancer in vitro modelling and drug screening. Journal of cancer research and clinical oncology 2019, 145, 2637-2647, doi:10.1007/s00432-019-03004-z.
103. Seidlitz, T.; Merker, S.R.; Rothe, A.; Zakrzewski, F.; von Neubeck, C.; Grützmann, K.; Sommer, U.; Schweitzer, C.; Schölch, S.; Uhlemann, H.; et al. Human gastric cancer modelling using organoids. Gut 2019, 68, 207-217, doi:10.1136/gutjnl-2017-314549.
104. Giobbe, G.G.; Bonfante, F.; Jones, B.C.; Gagliano, O.; Luni, C.; Zambaiti, E.; Perin, S.; Laterza, C.; Busslinger, G.; Stuart, H.; et al. SARS-CoV-2 infection and replication in human gastric organoids. Nature communications 2021, 12, 6610, doi:10.1038/s41467-021-26762-2.
105. Vlachogiannis, G.; Hedayat, S.; Vatsiou, A.; Jamin, Y.; Fernández-Mateos, J.; Khan, K.; Lampis, A.; Eason, K.; Huntingford, I.; Burke, R.; et al. Patient-derived organoids model treatment response of metastatic gastrointestinal cancers. Science (New York, N.Y.) 2018, 359, 920-926, doi:10.1126/science.aao2774.
67. Nanki, K.; Toshimitsu, K.; Takano, A.; Fujii, M.; Shimokawa, M.; Ohta, Y.; Matano, M.; Seino, T.; Nishikori, S.; Ishikawa, K.; et al. Divergent Routes toward Wnt and R-spondin Niche Independency during Human Gastric Carcinogenesis. Cell 2018, 174, 856-869.e817, doi:10.1016/j.cell.2018.07.027.
65. Steele, N.G.; Chakrabarti, J.; Wang, J.; Biesiada, J.; Holokai, L.; Chang, J.; Nowacki, L.M.; Hawkins, J.; Mahe, M.; Sundaram, N.; et al. An Organoid-Based Preclinical Model of Human Gastric Cancer. Cellular and molecular gastroenterology and hepatology 2019, 7, 161-184, doi:10.1016/j.jcmgh.2018.09.008.
106. Gobert, A.P.; Boutaud, O.; Asim, M.; Zagol-Ikapitte, I.A.; Delgado, A.G.; Latour, Y.L.; Finley, J.L.; Singh, K.; Verriere, T.G.; Allaman, M.M.; et al. Dicarbonyl Electrophiles Mediate Inflammation-Induced Gastrointestinal Carcinogenesis. Gastroenterology 2021, 160, 1256-1268.e1259, doi:10.1053/j.gastro.2020.11.006.
73. Eicher, A.K.; Kechele, D.O.; Sundaram, N.; Berns, H.M.; Poling, H.M.; Haines, L.E.; Sanchez, J.G.; Kishimoto, K.; Krishnamurthy, M.; Han, L.; et al. Functional human gastrointestinal organoids can be engineered from three primary germ layers derived separately from pluripotent stem cells. Cell stem cell 2022, 29, 36-51.e36, doi:10.1016/j.stem.2021.10.010.
86. Sierra, J.C.; Piazuelo, M.B.; Luis, P.B.; Barry, D.P.; Allaman, M.M.; Asim, M.; Sebrell, T.A.; Finley, J.L.; Rose, K.L.; Hill, S.; et al. Spermine oxidase mediates Helicobacter pylori-induced gastric inflammation, DNA damage, and carcinogenic signaling. Oncogene 2020, 39, 4465-4474, doi:10.1038/s41388-020-1304-6.
107. Koh, V.; Chakrabarti, J.; Torvund, M.; Steele, N.; Hawkins, J.A.; Ito, Y.; Wang, J.; Helmrath, M.A.; Merchant, J.L.; Ahmed, S.A.; et al. Hedgehog transcriptional effector GLI mediates mTOR-Induced PD-L1 expression in gastric cancer organoids. Cancer letters 2021, 518, 59-71, doi:10.1016/j.canlet.2021.06.007.
108. Togasaki, K.; Sugimoto, S.; Ohta, Y.; Nanki, K.; Matano, M.; Takahashi, S.; Fujii, M.; Kanai, T.; Sato, T. Wnt Signaling Shapes the Histologic Variation in Diffuse Gastric Cancer. Gastroenterology 2021, 160, 823-830, doi:10.1053/j.gastro.2020.10.047.
109. Chakrabarti, J.; Koh, V.; So, J.B.Y.; Yong, W.P.; Zavros, Y. A Preclinical Human-Derived Autologous Gastric Cancer Organoid/Immune Cell Co-Culture Model to Predict the Efficacy of Targeted Therapies. Journal of visualized experiments : JoVE 2021, doi:10.3791/61443.
109. Chakrabarti, J.; Koh, V.; So, J.B.Y.; Yong, W.P.; Zavros, Y. A Preclinical Human-Derived Autologous Gastric Cancer Organoid/Immune Cell Co-Culture Model to Predict the Efficacy of Targeted Therapies. Journal of visualized experiments : JoVE 2021, doi:10.3791/61443.
66. Lo, Y.H.; Kolahi, K.S.; Du, Y.; Chang, C.Y.; Krokhotin, A.; Nair, A.; Sobba, W.D.; Karlsson, K.; Jones, S.J.; Longacre, T.A.; et al. A CRISPR/Cas9-Engineered ARID1A-Deficient Human Gastric Cancer Organoid Model Reveals Essential and Nonessential Modes of Oncogenic Transformation. Cancer discovery 2021, 11, 1562-1581, doi:10.1158/2159-8290.Cd-20-1109.
110. Miao, X.; Wang, C.; Chai, C.; Tang, H.; Hu, J.; Zhao, Z.; Luo, W.; Zhang, H.; Zhu, K.; Zhou, W.; et al. Establishment of gastric cancer organoid and its application in individualized therapy. Oncology letters 2022, 24, 447, doi:10.3892/ol.2022.13567.
69. Li, G.; Ma, S.; Wu, Q.; Kong, D.; Yang, Z.; Gu, Z.; Feng, L.; Zhang, K.; Cheng, S.; Tian, Y.; et al. Establishment of gastric signet ring cell carcinoma organoid for the therapeutic drug testing. Cell death discovery 2022, 8, 6, doi:10.1038/s41420-021-00803-7.
111. Yoon, C.; Lu, J.; Kim, B.J.; Cho, S.J.; Kim, J.H.; Moy, R.H.; Ryeom, S.W.; Yoon, S.S. Patient-Derived Organoids from Locally Advanced Gastric Adenocarcinomas Can Predict Resistance to Neoadjuvant Chemotherapy. Journal of gastrointestinal surgery : official journal of the Society for Surgery of the Alimentary Tract 2023, 27, 666-676, doi:10.1007/s11605-022-05568-7.

Reviewer 4 Report
This manuscript is well written and generally easy to read. However, the information about the use of patient derived gastric organoids for improving clinical outcomes for patients with stomach diseases is limited. If the Review could be updated by a more detailed, critical discussion of gastric organoid papers or more recent reports (see below) 1-10, it would be useful to readers. The authors need to make it clearer which articles they refer to are directly related to gastric cancer (e.g. several of the references between 86-92 are only indirectly related to gastric cancer, i.e. they are about brain and prostate cancers).
Overall this review provides a summary of the literature relating to gastric organoid culture, however, the inclusion of a discussion of more recent publications would strengthen the manuscript.
A selection of more recent publications
1. Gao M, Lin M, Rao M, Thompson H, Hirai K, Choi M, Georgakis GV, Sasson AR, Bucobo JC, Tzimas D. Development of patient-derived gastric cancer organoids from endoscopic biopsies and surgical tissues. Annals of surgical oncology 2018;25: 2767-75.
2. Li G, Ma S, Wu Q, Kong D, Yang Z, Gu Z, Feng L, Zhang K, Cheng S, Tian Y. Establishment of gastric signet ring cell carcinoma organoid for the therapeutic drug testing. Cell Death Discovery 2022;8: 6.
3. Mircetic J, Camgöz A, Abohawya M, Ding L, Dietzel J, Tobar SG, Paszkowski‐Rogacz M, Seidlitz T, Schmäche T, Mehnert MC. CRISPR/Cas9 Screen in Gastric Cancer Patient‐Derived Organoids Reveals KDM1A‐NDRG1 Axis as a Targetable Vulnerability. Small Methods 2023: 2201605.
4. Patil S, Jahagirdar S, Khot M, Sengupta K. Studying the role of chromosomal instability (CIN) in GI cancers using patient-derived organoids. Journal of Molecular Biology 2022;434: 167256.
5. Skubleny D, Purich K, Williams T, Wickware J, Ghosh S, Spratlin JL, Schiller DE, Rayat G. A 107-gene Nanostring assay effectively characterizes complex multiomic gastric cancer molecular classification in a translational patient-derived organoid model: American Society of Clinical Oncology, 2022.
6. Yang R, Yu Y. Patient-derived organoids in translational oncology and drug screening. Cancer Letters 2023: 216180.
7. Yoon C, Lu J, Kim B-J, Cho S-J, Kim JH, Moy RH, Ryeom SW, Yoon SS. Patient-Derived Organoids from Locally Advanced Gastric Adenocarcinomas Can Predict Resistance to Neoadjuvant Chemotherapy. Journal of Gastrointestinal Surgery 2023: 1-11.
8. Zhang H, Qin Y, Jia M, Li L, Zhang W, Li L, Zhang Z, Liu Y. A gastric cancer patient-derived three-dimensional cell spheroid culture model. American Journal of Cancer Research 2023;13: 964.
9. Zhao Y, Huang M, Zhu Y, Xu L, Li X, Yu J. IDDF2022-ABS-0105 Patient-derived organoids can predict chemotherapy response of gastric cancers and analysis of its molecular characteristics: BMJ Publishing Group, 2022.
10. Zou J, Wang S, Chai N, Yue H, Ye P, Guo P, Li F, Wei B, Ma G, Wei W. Construction of gastric cancer patient-derived organoids and their utilization in a comparative study of clinically used paclitaxel nanoformulations. Journal of Nanobiotechnology 2022;20: 1-16.
Author Response
Thank you for your careful review. We really appreciate your efforts in reviewing our manuscript. Your careful review has helped to make our study clearer and more comprehensive. Attached is the manuscript of our revised model.
Comment 1: Overall this review provides a summary of the literature relating to gastric organoid culture, however, the inclusion of a discussion of more recent publications would strengthen the manuscript.
Response 1: Thank you for your valuable suggestions, the list of articles you gave us has helped us a lot to modify our articles. We have revised the section "4.1 Tissue engineering" by removing some contents that are not directly related to gastric organoids and adding the application of gastric organoids in gene editing. Also, we rewrote the section “4.3 Drug testing and development”, updated the references and summarized the important articles related to gastric organoids in the last five years in Table 2, which we hope will be helpful for future readers. We have revised this part of the article as follows:
4.3 Drug testing and development
Gastric organoid is a reliable model for testing drugs. Anti-tumor drugs for gastric cancer can be classified into chemotherapeutic drugs and targeted drugs. Chemotherapy drugs are the most basic and important part of treatment, such as cisplatin, 5-fluorouracil (5-FU), irinotecan, and oxaliplatin. It has been found that the IC50 of various chemotherapeutic drugs tested in PDO is consistent with the results obtained in many clinical pharmacology studies [15,69], that is, the same drug can achieve consistent pharmacological effects in organoid models and in vivo. Similarly, various targeted drugs show the same effects in PDO [90]. The reason for this may be because gastric organoids can maintain a large proportion of the genetic characteristics of cancer tissues, such as various mutated genes. Researchers performed whole-exome sequencing (WES) analysis on 12 groups of gastric organoid models and tumor tissues, and found highly similar various mutated genes between them. In particular, within the mutated genes, the proportion of different base mutations was significantly similar between the two groups [69]. Therefore, PDO, as a biological model, can be trust for drug testing in vitro. In the recent five years, many researches used gastric organoids for drug testing (Table 2).
Gastric organoids are more efficient in drug testing and development such as detecting drug toxicity, developing new drugs and high-throughput screening [84]. Traditional screening of anti-tumor drugs is a complex process, whereas drug screening using PDO model can address some of the drawbacks of traditional methods. The traditional method is often based on the construction of animal models, so the screening process is more time-consuming and expensive, and the experimental cycle is longer. PDO models are derived from the patient, and more beneficial for individualized treatment. The 3D cell culture mode has a higher throughput, allowing for the screening of multiple drugs at the same time [91]. However, most of the current studies are still validating the reliability of PDO, in which new drug development is less reported.
Table 2. Summary of papers on gastric organoids in the last five years
|
Authors |
Tissue acquisition |
Stem cells |
Methods of isolating glands |
Applications |
|
Gao M et al. [15] |
Endoscope Surgery |
ASC |
Physical pressure |
Next-generation sequencing Drugs testing |
|
Li J et al.[102] |
Ascites puncture |
ASC |
- |
Malignant ascites-derived organoid Drugs testing |
|
Seidlitz T et al.[103] |
Surgery |
ASC |
Collagenase XI Dispase II |
Whole genome sequencing RNA sequencing Drugs testing Targeted therapy testing Biobank establishment |
|
Giobbe GG. et al.[104] |
Biopsy |
ASC |
Physical pressure |
SARS-CoV-2 infection |
|
Vlachogiannis G et al.[105] |
Biopsy |
ASC |
TrypLE |
Whole genome sequencing Drugs testing Targeted therapy testing Biobank establishment |
|
Nanki K et al.[67] |
Surgery Endoscope Ascites puncture |
ASC |
Libase TH TrypLE |
CRISPR/Cas9 Transcriptomic analysis Xenotransplantation of Organoids Biobank establishment |
|
Steele NG et al.[65] |
Surgery |
ASC |
Collagenase Hyaluronidase |
RNA sequencing Drugs testing Targeted therapy testing Organoids tumorigenicity analysis |
|
Gobert AP et al.[106] |
Endoscope |
ASC |
- |
H. pylori |
|
Eicher AK et al.[73] |
- |
hPSC |
- |
Organoid assembly approach Tissue engineering |
|
Sierra JC et al.[86] |
Surgery |
ASC |
Collagenase IV |
H. pylori |
|
Koh V et al.[107] |
Surgery |
ASC |
Collagenase |
Immunoregulation in EMT |
|
Togasaki K et al. [108] |
Surgery Ascites puncture |
ASC |
Libase TH |
Whole-exome Sequence RNA Sequence Xenotransplantation of Organoids |
|
Chakrabarti J et al. [109] |
Biopsy |
ASC |
Collagenase I Hyaluronidase IV-S |
Organoid/immune cells co-culture model Targeted therapy testing |
|
Lo YH et al.[66] |
Surgery |
ASC |
Collagenase I |
CRISPR/Cas9 |
|
Miao X et al.[110] |
Surgery |
ASC |
Collagenase II Dispase II |
Drugs testing |
|
Li G et al.[69] |
Surgery |
ASC |
Collagenase |
Organoids tumorigenicity analysis Drugs testing |
|
Yoon C et al.[111] |
Endoscope |
ASC |
Collagenase III |
Drugs testing |
|
Zhang H et al.[90] |
Surgery |
ASC |
Gentle MACS™ Dissociator tumor dissociation kit |
Transcriptome sequencing Drugs testing |
15. Gao, M.; Lin, M.; Rao, M.; Thompson, H.; Hirai, K.; Choi, M.; Georgakis, G.V.; Sasson, A.R.; Bucobo, J.C.; Tzimas, D.; et al. Development of Patient-Derived Gastric Cancer Organoids from Endoscopic Biopsies and Surgical Tissues. Annals of surgical oncology 2018, 25, 2767-2775, doi:10.1245/s10434-018-6662-8.
102. Li, J.; Xu, H.; Zhang, L.; Song, L.; Feng, D.; Peng, X.; Wu, M.; Zou, Y.; Wang, B.; Zhan, L.; et al. Malignant ascites-derived organoid (MADO) cultures for gastric cancer in vitro modelling and drug screening. Journal of cancer research and clinical oncology 2019, 145, 2637-2647, doi:10.1007/s00432-019-03004-z.
103. Seidlitz, T.; Merker, S.R.; Rothe, A.; Zakrzewski, F.; von Neubeck, C.; Grützmann, K.; Sommer, U.; Schweitzer, C.; Schölch, S.; Uhlemann, H.; et al. Human gastric cancer modelling using organoids. Gut 2019, 68, 207-217, doi:10.1136/gutjnl-2017-314549.
104. Giobbe, G.G.; Bonfante, F.; Jones, B.C.; Gagliano, O.; Luni, C.; Zambaiti, E.; Perin, S.; Laterza, C.; Busslinger, G.; Stuart, H.; et al. SARS-CoV-2 infection and replication in human gastric organoids. Nature communications 2021, 12, 6610, doi:10.1038/s41467-021-26762-2.
105. Vlachogiannis, G.; Hedayat, S.; Vatsiou, A.; Jamin, Y.; Fernández-Mateos, J.; Khan, K.; Lampis, A.; Eason, K.; Huntingford, I.; Burke, R.; et al. Patient-derived organoids model treatment response of metastatic gastrointestinal cancers. Science (New York, N.Y.) 2018, 359, 920-926, doi:10.1126/science.aao2774.
67. Nanki, K.; Toshimitsu, K.; Takano, A.; Fujii, M.; Shimokawa, M.; Ohta, Y.; Matano, M.; Seino, T.; Nishikori, S.; Ishikawa, K.; et al. Divergent Routes toward Wnt and R-spondin Niche Independency during Human Gastric Carcinogenesis. Cell 2018, 174, 856-869.e817, doi:10.1016/j.cell.2018.07.027.
65. Steele, N.G.; Chakrabarti, J.; Wang, J.; Biesiada, J.; Holokai, L.; Chang, J.; Nowacki, L.M.; Hawkins, J.; Mahe, M.; Sundaram, N.; et al. An Organoid-Based Preclinical Model of Human Gastric Cancer. Cellular and molecular gastroenterology and hepatology 2019, 7, 161-184, doi:10.1016/j.jcmgh.2018.09.008.
106. Gobert, A.P.; Boutaud, O.; Asim, M.; Zagol-Ikapitte, I.A.; Delgado, A.G.; Latour, Y.L.; Finley, J.L.; Singh, K.; Verriere, T.G.; Allaman, M.M.; et al. Dicarbonyl Electrophiles Mediate Inflammation-Induced Gastrointestinal Carcinogenesis. Gastroenterology 2021, 160, 1256-1268.e1259, doi:10.1053/j.gastro.2020.11.006.
73. Eicher, A.K.; Kechele, D.O.; Sundaram, N.; Berns, H.M.; Poling, H.M.; Haines, L.E.; Sanchez, J.G.; Kishimoto, K.; Krishnamurthy, M.; Han, L.; et al. Functional human gastrointestinal organoids can be engineered from three primary germ layers derived separately from pluripotent stem cells. Cell stem cell 2022, 29, 36-51.e36, doi:10.1016/j.stem.2021.10.010.
86. Sierra, J.C.; Piazuelo, M.B.; Luis, P.B.; Barry, D.P.; Allaman, M.M.; Asim, M.; Sebrell, T.A.; Finley, J.L.; Rose, K.L.; Hill, S.; et al. Spermine oxidase mediates Helicobacter pylori-induced gastric inflammation, DNA damage, and carcinogenic signaling. Oncogene 2020, 39, 4465-4474, doi:10.1038/s41388-020-1304-6.
107. Koh, V.; Chakrabarti, J.; Torvund, M.; Steele, N.; Hawkins, J.A.; Ito, Y.; Wang, J.; Helmrath, M.A.; Merchant, J.L.; Ahmed, S.A.; et al. Hedgehog transcriptional effector GLI mediates mTOR-Induced PD-L1 expression in gastric cancer organoids. Cancer letters 2021, 518, 59-71, doi:10.1016/j.canlet.2021.06.007.
108. Togasaki, K.; Sugimoto, S.; Ohta, Y.; Nanki, K.; Matano, M.; Takahashi, S.; Fujii, M.; Kanai, T.; Sato, T. Wnt Signaling Shapes the Histologic Variation in Diffuse Gastric Cancer. Gastroenterology 2021, 160, 823-830, doi:10.1053/j.gastro.2020.10.047.
109. Chakrabarti, J.; Koh, V.; So, J.B.Y.; Yong, W.P.; Zavros, Y. A Preclinical Human-Derived Autologous Gastric Cancer Organoid/Immune Cell Co-Culture Model to Predict the Efficacy of Targeted Therapies. Journal of visualized experiments : JoVE 2021, doi:10.3791/61443.
109. Chakrabarti, J.; Koh, V.; So, J.B.Y.; Yong, W.P.; Zavros, Y. A Preclinical Human-Derived Autologous Gastric Cancer Organoid/Immune Cell Co-Culture Model to Predict the Efficacy of Targeted Therapies. Journal of visualized experiments : JoVE 2021, doi:10.3791/61443.
66. Lo, Y.H.; Kolahi, K.S.; Du, Y.; Chang, C.Y.; Krokhotin, A.; Nair, A.; Sobba, W.D.; Karlsson, K.; Jones, S.J.; Longacre, T.A.; et al. A CRISPR/Cas9-Engineered ARID1A-Deficient Human Gastric Cancer Organoid Model Reveals Essential and Nonessential Modes of Oncogenic Transformation. Cancer discovery 2021, 11, 1562-1581, doi:10.1158/2159-8290.Cd-20-1109.
110. Miao, X.; Wang, C.; Chai, C.; Tang, H.; Hu, J.; Zhao, Z.; Luo, W.; Zhang, H.; Zhu, K.; Zhou, W.; et al. Establishment of gastric cancer organoid and its application in individualized therapy. Oncology letters 2022, 24, 447, doi:10.3892/ol.2022.13567.
69. Li, G.; Ma, S.; Wu, Q.; Kong, D.; Yang, Z.; Gu, Z.; Feng, L.; Zhang, K.; Cheng, S.; Tian, Y.; et al. Establishment of gastric signet ring cell carcinoma organoid for the therapeutic drug testing. Cell death discovery 2022, 8, 6, doi:10.1038/s41420-021-00803-7.
111. Yoon, C.; Lu, J.; Kim, B.J.; Cho, S.J.; Kim, J.H.; Moy, R.H.; Ryeom, S.W.; Yoon, S.S. Patient-Derived Organoids from Locally Advanced Gastric Adenocarcinomas Can Predict Resistance to Neoadjuvant Chemotherapy. Journal of gastrointestinal surgery : official journal of the Society for Surgery of the Alimentary Tract 2023, 27, 666-676, doi:10.1007/s11605-022-05568-7.

Reviewer 5 Report
In this paper the authors aimed firstly to review the current literature on the establishment of organoid cultures, and secondly to explore organoid translational applications. The manuscript is well written however it is not innovative, it lacks some relevant references, for example Establishment of gastric cancer organoid and its application in individualized therapy. Miao X, et al Oncol Lett. 2022. It would be useful create a summary table of the most important papers as well as it would be useful a diagram or a graphic summary to accompany the single image contained in the manuscript which should help technicians and researchers in choosing the most appropriate method for the creation of the organoid. The manuscript is not ready for publication.Author Response
Thank you for your careful review. We really appreciate your efforts in reviewing our manuscript. Your careful review has helped to make our study clearer and more comprehensive. Attached is the manuscript of our revised model.
Comment 1: It is not innovative. It lacks some relevant references, for example Establishment of gastric cancer organoid and its application in individualized therapy.
Response 1: Thank you for your valuable suggestions. Thank you for giving me the idea of individualized treatment. The following changes were made to the manuscript to respond to this idea: First, Screening drugs is an important part of individualized therapy, so we rewrote the section ‘4.3 Drug testing and development’. We present the reliability of the gastric organoid. As a still novel biological model, it is essential to verify the reliability. Next, we review the advantages of gastric organoid for screening drug. We have updated the most recent papers and focused on the application of gastric organoids. We have revised this part of the article as follows:
4.3 Drug testing and development
Gastric organoid is a reliable model for testing drugs. Anti-tumor drugs for gastric cancer can be classified into chemotherapeutic drugs and targeted drugs. Chemotherapy drugs are the most basic and important part of treatment, such as cisplatin, 5-fluorouracil (5-FU), irinotecan, and oxaliplatin. It has been found that the IC50 of various chemotherapeutic drugs tested in PDO is consistent with the results obtained in many clinical pharmacology studies [15,69], that is, the same drug can achieve consistent pharmacological effects in organoid models and in vivo. Similarly, various targeted drugs show the same effects in PDO[90]. The reason for this may be because gastric organoids can maintain a large proportion of the genetic characteristics of cancer tissues, such as various mutated genes. Researchers performed whole-exome sequencing (WES) analysis on 12 groups of gastric organoid models and tumor tissues, and found highly similar various mutated genes between them. In particular, within the mutated genes, the proportion of different base mutations was significantly similar between the two groups [69]. Therefore, PDO, as a biological model, can be trust for drug testing in vitro. In the recent five years, many researches used gastric organoids for drug testing (Table 2).
Gastric organoids are more efficient in drug testing and development such as detecting drug toxicity, developing new drugs and high-throughput screening [84]. Traditional screening of anti-tumor drugs is a complex process, whereas drug screening using PDO model can address some of the drawbacks of traditional methods. The traditional method is often based on the construction of animal models, so the screening process is more time-consuming and expensive, and the experimental cycle is longer. PDO models are derived from the patient, and more beneficial for individualized treatment. The 3D cell culture mode has a higher throughput, allowing for the screening of multiple drugs at the same time [91]. However, most of the current studies are still validating the reliability of PDO, in which new drug development is less reported.
Comment 2: It would be useful create a summary table of the most important papers as well as it would be useful a diagram or a graphic summary to accompany the single image contained in the manuscript.
Response 2: Thank you for your precious advice. We have updated the references and summarized the important articles related to gastric organoids in the last five years in Table 2. We list not only the main applications of the gastric organoid, but also the controversies and different methods between some of the reported studies, such as the method of gland isolation and the means of tissue acquisition, which we hope will be helpful for future readers.
Table 2. Summary of papers on gastric organoids in the last five years
|
Authors |
Tissue acquisition |
Stem cells |
Methods of isolating glands |
Applications |
|
Gao M et al. [15] |
Endoscope Surgery |
ASC |
Physical pressure |
Next-generation sequencing Drugs testing |
|
Li J et al.[102] |
Ascites puncture |
ASC |
- |
Malignant ascites-derived organoid Drugs testing |
|
Seidlitz T et al.[103] |
Surgery |
ASC |
Collagenase XI Dispase II |
Whole genome sequencing RNA sequencing Drugs testing Targeted therapy testing Biobank establishment |
|
Giobbe GG. et al.[104] |
Biopsy |
ASC |
Physical pressure |
SARS-CoV-2 infection |
|
Vlachogiannis G et al.[105] |
Biopsy |
ASC |
TrypLE |
Whole genome sequencing Drugs testing Targeted therapy testing Biobank establishment |
|
Nanki K et al.[67] |
Surgery Endoscope Ascites puncture |
ASC |
Libase TH TrypLE |
CRISPR/Cas9 Transcriptomic analysis Xenotransplantation of Organoids Biobank establishment |
|
Steele NG et al.[65] |
Surgery |
ASC |
Collagenase Hyaluronidase |
RNA sequencing Drugs testing Targeted therapy testing Organoids tumorigenicity analysis |
|
Gobert AP et al.[106] |
Endoscope |
ASC |
- |
H. pylori |
|
Eicher AK et al.[73] |
- |
hPSC |
- |
Organoid assembly approach Tissue engineering |
|
Sierra JC et al.[86] |
Surgery |
ASC |
Collagenase IV |
H. pylori |
|
Koh V et al.[107] |
Surgery |
ASC |
Collagenase |
Immunoregulation in EMT |
|
Togasaki K et al. [108] |
Surgery Ascites puncture |
ASC |
Libase TH |
Whole-exome Sequence RNA Sequence Xenotransplantation of Organoids |
|
Chakrabarti J et al. [109] |
Biopsy |
ASC |
Collagenase I Hyaluronidase IV-S |
Organoid/immune cells co-culture model Targeted therapy testing |
|
Lo YH et al.[66] |
Surgery |
ASC |
Collagenase I |
CRISPR/Cas9 |
|
Miao X et al.[110] |
Surgery |
ASC |
Collagenase II Dispase II |
Drugs testing |
|
Li G et al.[69] |
Surgery |
ASC |
Collagenase |
Organoids tumorigenicity analysis Drugs testing |
|
Yoon C et al.[111] |
Endoscope |
ASC |
Collagenase III |
Drugs testing |
|
Zhang H et al.[90] |
Surgery |
ASC |
Gentle MACS™ Dissociator tumor dissociation kit |
Transcriptome sequencing Drugs testing |
91. Seidlitz, T.; Koo, B.K.; Stange, D.E. Gastric organoids-an in vitro model system for the study of gastric development and road to personalized medicine. Cell death and differentiation 2021, 28, 68-83, doi:10.1038/s41418-020-00662-2.
15. Gao, M.; Lin, M.; Rao, M.; Thompson, H.; Hirai, K.; Choi, M.; Georgakis, G.V.; Sasson, A.R.; Bucobo, J.C.; Tzimas, D.; et al. Development of Patient-Derived Gastric Cancer Organoids from Endoscopic Biopsies and Surgical Tissues. Annals of surgical oncology 2018, 25, 2767-2775, doi:10.1245/s10434-018-6662-8.
102. Li, J.; Xu, H.; Zhang, L.; Song, L.; Feng, D.; Peng, X.; Wu, M.; Zou, Y.; Wang, B.; Zhan, L.; et al. Malignant ascites-derived organoid (MADO) cultures for gastric cancer in vitro modelling and drug screening. Journal of cancer research and clinical oncology 2019, 145, 2637-2647, doi:10.1007/s00432-019-03004-z.
103. Seidlitz, T.; Merker, S.R.; Rothe, A.; Zakrzewski, F.; von Neubeck, C.; Grützmann, K.; Sommer, U.; Schweitzer, C.; Schölch, S.; Uhlemann, H.; et al. Human gastric cancer modelling using organoids. Gut 2019, 68, 207-217, doi:10.1136/gutjnl-2017-314549.
104. Giobbe, G.G.; Bonfante, F.; Jones, B.C.; Gagliano, O.; Luni, C.; Zambaiti, E.; Perin, S.; Laterza, C.; Busslinger, G.; Stuart, H.; et al. SARS-CoV-2 infection and replication in human gastric organoids. Nature communications 2021, 12, 6610, doi:10.1038/s41467-021-26762-2.
105. Vlachogiannis, G.; Hedayat, S.; Vatsiou, A.; Jamin, Y.; Fernández-Mateos, J.; Khan, K.; Lampis, A.; Eason, K.; Huntingford, I.; Burke, R.; et al. Patient-derived organoids model treatment response of metastatic gastrointestinal cancers. Science (New York, N.Y.) 2018, 359, 920-926, doi:10.1126/science.aao2774.
67. Nanki, K.; Toshimitsu, K.; Takano, A.; Fujii, M.; Shimokawa, M.; Ohta, Y.; Matano, M.; Seino, T.; Nishikori, S.; Ishikawa, K.; et al. Divergent Routes toward Wnt and R-spondin Niche Independency during Human Gastric Carcinogenesis. Cell 2018, 174, 856-869.e817, doi:10.1016/j.cell.2018.07.027.
65. Steele, N.G.; Chakrabarti, J.; Wang, J.; Biesiada, J.; Holokai, L.; Chang, J.; Nowacki, L.M.; Hawkins, J.; Mahe, M.; Sundaram, N.; et al. An Organoid-Based Preclinical Model of Human Gastric Cancer. Cellular and molecular gastroenterology and hepatology 2019, 7, 161-184, doi:10.1016/j.jcmgh.2018.09.008.
106. Gobert, A.P.; Boutaud, O.; Asim, M.; Zagol-Ikapitte, I.A.; Delgado, A.G.; Latour, Y.L.; Finley, J.L.; Singh, K.; Verriere, T.G.; Allaman, M.M.; et al. Dicarbonyl Electrophiles Mediate Inflammation-Induced Gastrointestinal Carcinogenesis. Gastroenterology 2021, 160, 1256-1268.e1259, doi:10.1053/j.gastro.2020.11.006.
73. Eicher, A.K.; Kechele, D.O.; Sundaram, N.; Berns, H.M.; Poling, H.M.; Haines, L.E.; Sanchez, J.G.; Kishimoto, K.; Krishnamurthy, M.; Han, L.; et al. Functional human gastrointestinal organoids can be engineered from three primary germ layers derived separately from pluripotent stem cells. Cell stem cell 2022, 29, 36-51.e36, doi:10.1016/j.stem.2021.10.010.
86. Sierra, J.C.; Piazuelo, M.B.; Luis, P.B.; Barry, D.P.; Allaman, M.M.; Asim, M.; Sebrell, T.A.; Finley, J.L.; Rose, K.L.; Hill, S.; et al. Spermine oxidase mediates Helicobacter pylori-induced gastric inflammation, DNA damage, and carcinogenic signaling. Oncogene 2020, 39, 4465-4474, doi:10.1038/s41388-020-1304-6.
107. Koh, V.; Chakrabarti, J.; Torvund, M.; Steele, N.; Hawkins, J.A.; Ito, Y.; Wang, J.; Helmrath, M.A.; Merchant, J.L.; Ahmed, S.A.; et al. Hedgehog transcriptional effector GLI mediates mTOR-Induced PD-L1 expression in gastric cancer organoids. Cancer letters 2021, 518, 59-71, doi:10.1016/j.canlet.2021.06.007.
108. Togasaki, K.; Sugimoto, S.; Ohta, Y.; Nanki, K.; Matano, M.; Takahashi, S.; Fujii, M.; Kanai, T.; Sato, T. Wnt Signaling Shapes the Histologic Variation in Diffuse Gastric Cancer. Gastroenterology 2021, 160, 823-830, doi:10.1053/j.gastro.2020.10.047.
109. Chakrabarti, J.; Koh, V.; So, J.B.Y.; Yong, W.P.; Zavros, Y. A Preclinical Human-Derived Autologous Gastric Cancer Organoid/Immune Cell Co-Culture Model to Predict the Efficacy of Targeted Therapies. Journal of visualized experiments : JoVE 2021, doi:10.3791/61443.
109. Chakrabarti, J.; Koh, V.; So, J.B.Y.; Yong, W.P.; Zavros, Y. A Preclinical Human-Derived Autologous Gastric Cancer Organoid/Immune Cell Co-Culture Model to Predict the Efficacy of Targeted Therapies. Journal of visualized experiments : JoVE 2021, doi:10.3791/61443.
66. Lo, Y.H.; Kolahi, K.S.; Du, Y.; Chang, C.Y.; Krokhotin, A.; Nair, A.; Sobba, W.D.; Karlsson, K.; Jones, S.J.; Longacre, T.A.; et al. A CRISPR/Cas9-Engineered ARID1A-Deficient Human Gastric Cancer Organoid Model Reveals Essential and Nonessential Modes of Oncogenic Transformation. Cancer discovery 2021, 11, 1562-1581, doi:10.1158/2159-8290.Cd-20-1109.
110. Miao, X.; Wang, C.; Chai, C.; Tang, H.; Hu, J.; Zhao, Z.; Luo, W.; Zhang, H.; Zhu, K.; Zhou, W.; et al. Establishment of gastric cancer organoid and its application in individualized therapy. Oncology letters 2022, 24, 447, doi:10.3892/ol.2022.13567.
69. Li, G.; Ma, S.; Wu, Q.; Kong, D.; Yang, Z.; Gu, Z.; Feng, L.; Zhang, K.; Cheng, S.; Tian, Y.; et al. Establishment of gastric signet ring cell carcinoma organoid for the therapeutic drug testing. Cell death discovery 2022, 8, 6, doi:10.1038/s41420-021-00803-7.
111. Yoon, C.; Lu, J.; Kim, B.J.; Cho, S.J.; Kim, J.H.; Moy, R.H.; Ryeom, S.W.; Yoon, S.S. Patient-Derived Organoids from Locally Advanced Gastric Adenocarcinomas Can Predict Resistance to Neoadjuvant Chemotherapy. Journal of gastrointestinal surgery : official journal of the Society for Surgery of the Alimentary Tract 2023, 27, 666-676, doi:10.1007/s11605-022-05568-7.

Round 2
Reviewer 2 Report
I think the auhors have revised the manuscript accordig to reviewers suggestions.We can accept the papaer in the current form.
Author Response
We really appreciate your efforts in reviewing our manuscript. Your comments have helped our manuscript more readable and clearer.

Reviewer 3 Report
The review can be accepted.
Author Response

(The authors gave the same response as above.)

Reviewer 4 Report
The authors responses have improved the manuscript - there is more up to date information and the focus on the use of gastric organoids haas increased. The Table listing key references is fine, but I would include the year of each publication in the Table. The English in the new sections is mainly fine, but careful, editorial proof reading is required.
A little more in depth discussion of the results might help the readers, but the manuscript contains sufficient information to be useful.
Author Response
We really appreciate your efforts in reviewing our manuscript. Your comments have helped our manuscript more readable and clearer. We have modified this manuscript according to your requirements, as follows:
Comment 1: The Table listing key references is fine, but I would include the year of each publication in the Table.
Response 1: Thank for your comments. We have added the publication dates of the papers within Table 2 and rearranged them in chronological order. Thank you again for your contribution to our paper.
Table 2. Summary of studies on gastric organoids in the last five years
|
Year |
Authors |
Tissue acquisition |
Stem cells |
Methods of isolating glands |
Applications |
|
2018 |
Gao M et al. [15] |
Endoscope Surgery |
ASC |
Physical pressure |
Next-generation sequencing Drugs testing |
|
2018 |
Vlachogiannis G et al.[102] |
Biopsy |
ASC |
TrypLE |
Whole genome sequencing Drugs testing Targeted therapy testing Biobank establishment |
|
2018 |
Nanki K et al.[67] |
Surgery Endoscope Ascites puncture |
ASC |
Libase TH TrypLE |
CRISPR/Cas9 Transcriptomic analysis Xenotransplantation of Organoids Biobank establishment |
|
2019 |
Steele NG et al.[65] |
Surgery |
ASC |
Collagenase Hyaluronidase |
RNA sequencing Drugs testing Targeted therapy testing Organoids tumorigenicity analysis |
|
2019 |
Li J et al.[103] |
Ascites puncture |
ASC |
- |
Malignant ascites-derived organoid Drugs testing |
|
2019 |
Seidlitz T et al.[104] |
Surgery |
ASC |
Collagenase XI Dispase II |
Whole genome sequencing RNA sequencing Drugs testing Targeted therapy testing Biobank establishment |
|
2020 |
Sierra JC et al.[86] |
Surgery |
ASC |
Collagenase IV |
H. pylori |
|
2021 |
Giobbe GG. et al.[105] |
Biopsy |
ASC |
Physical pressure |
SARS-CoV-2 infection |
|
2021 |
Gobert AP et al.[106] |
Endoscope |
ASC |
- |
H. pylori |
|
2021 |
Koh V et al.[107] |
Surgery |
ASC |
Collagenase |
Immunoregulation in EMT |
|
2021 |
Togasaki K et al. [108] |
Surgery Ascites puncture |
ASC |
Libase TH |
Whole-exome Sequence RNA Sequence Xenotransplantation of Organoids |
|
2021 |
Chakrabarti J et al. [109] |
Biopsy |
ASC |
Collagenase I Hyaluronidase IV-S |
Organoid/immune cells co-culture model Targeted therapy testing |
|
2021 |
Lo YH et al.[66] |
Surgery |
ASC |
Collagenase I |
CRISPR/Cas9 |
|
2022 |
Eicher AK et al.[73] |
- |
hPSC |
- |
Organoid assembly approach Tissue engineering |
|
2022 |
Miao X et al.[110] |
Surgery |
ASC |
Collagenase II Dispase II |
Drugs testing |
|
2022 |
Li G et al.[69] |
Surgery |
ASC |
Collagenase |
Organoids tumorigenicity analysis Drugs testing |
|
2023 |
Yoon C et al.[111] |
Endoscope |
ASC |
Collagenase III |
Drugs testing |
|
2023 |
Zhang H et al.[90] |
Surgery |
ASC |
Gentle MACS™ Dissociator tumor dissociation kit |
Transcriptome sequencing Drugs testing |
15. Gao, M.; Lin, M.; Rao, M.; Thompson, H.; Hirai, K.; Choi, M.; Georgakis, G.V.; Sasson, A.R.; Bucobo, J.C.; Tzimas, D.; et al. Development of Patient-Derived Gastric Cancer Organoids from Endoscopic Biopsies and Surgical Tissues. Annals of surgical oncology 2018, 25, 2767-2775, doi:10.1245/s10434-018-6662-8.
102. Vlachogiannis, G.; Hedayat, S.; Vatsiou, A.; Jamin, Y.; Fernández-Mateos, J.; Khan, K.; Lampis, A.; Eason, K.; Huntingford, I.; Burke, R.; et al. Patient-derived organoids model treatment response of metastatic gastrointestinal cancers. Science (New York, N.Y.) 2018, 359, 920-926, doi:10.1126/science.aao2774.
67. Nanki, K.; Toshimitsu, K.; Takano, A.; Fujii, M.; Shimokawa, M.; Ohta, Y.; Matano, M.; Seino, T.; Nishikori, S.; Ishikawa, K.; et al. Divergent Routes toward Wnt and R-spondin Niche Independency during Human Gastric Carcinogenesis. Cell 2018, 174, 856-869.e817, doi:10.1016/j.cell.2018.07.027.
65. Steele, N.G.; Chakrabarti, J.; Wang, J.; Biesiada, J.; Holokai, L.; Chang, J.; Nowacki, L.M.; Hawkins, J.; Mahe, M.; Sundaram, N.; et al. An Organoid-Based Preclinical Model of Human Gastric Cancer. Cellular and molecular gastroenterology and hepatology 2019, 7, 161-184, doi:10.1016/j.jcmgh.2018.09.008.
103. Li, J.; Xu, H.; Zhang, L.; Song, L.; Feng, D.; Peng, X.; Wu, M.; Zou, Y.; Wang, B.; Zhan, L.; et al. Malignant ascites-derived organoid (MADO) cultures for gastric cancer in vitro modelling and drug screening. Journal of cancer research and clinical oncology 2019, 145, 2637-2647, doi:10.1007/s00432-019-03004-z.
104. Seidlitz, T.; Merker, S.R.; Rothe, A.; Zakrzewski, F.; von Neubeck, C.; Grützmann, K.; Sommer, U.; Schweitzer, C.; Schölch, S.; Uhlemann, H.; et al. Human gastric cancer modelling using organoids. Gut 2019, 68, 207-217, doi:10.1136/gutjnl-2017-314549.
86. Sierra, J.C.; Piazuelo, M.B.; Luis, P.B.; Barry, D.P.; Allaman, M.M.; Asim, M.; Sebrell, T.A.; Finley, J.L.; Rose, K.L.; Hill, S.; et al. Spermine oxidase mediates Helicobacter pylori-induced gastric inflammation, DNA damage, and carcinogenic signaling. Oncogene 2020, 39, 4465-4474, doi:10.1038/s41388-020-1304-6.
105. Giobbe, G.G.; Bonfante, F.; Jones, B.C.; Gagliano, O.; Luni, C.; Zambaiti, E.; Perin, S.; Laterza, C.; Busslinger, G.; Stuart, H.; et al. SARS-CoV-2 infection and replication in human gastric organoids. Nature communications 2021, 12, 6610, doi:10.1038/s41467-021-26762-2.
106. Gobert, A.P.; Boutaud, O.; Asim, M.; Zagol-Ikapitte, I.A.; Delgado, A.G.; Latour, Y.L.; Finley, J.L.; Singh, K.; Verriere, T.G.; Allaman, M.M.; et al. Dicarbonyl Electrophiles Mediate Inflammation-Induced Gastrointestinal Carcinogenesis. Gastroenterology 2021, 160, 1256-1268.e1259, doi:10.1053/j.gastro.2020.11.006.
107. Koh, V.; Chakrabarti, J.; Torvund, M.; Steele, N.; Hawkins, J.A.; Ito, Y.; Wang, J.; Helmrath, M.A.; Merchant, J.L.; Ahmed, S.A.; et al. Hedgehog transcriptional effector GLI mediates mTOR-Induced PD-L1 expression in gastric cancer organoids. Cancer letters 2021, 518, 59-71, doi:10.1016/j.canlet.2021.06.007.
108. Togasaki, K.; Sugimoto, S.; Ohta, Y.; Nanki, K.; Matano, M.; Takahashi, S.; Fujii, M.; Kanai, T.; Sato, T. Wnt Signaling Shapes the Histologic Variation in Diffuse Gastric Cancer. Gastroenterology 2021, 160, 823-830, doi:10.1053/j.gastro.2020.10.047.
109. Chakrabarti, J.; Koh, V.; So, J.B.Y.; Yong, W.P.; Zavros, Y. A Preclinical Human-Derived Autologous Gastric Cancer Organoid/Immune Cell Co-Culture Model to Predict the Efficacy of Targeted Therapies. Journal of visualized experiments : JoVE 2021, doi:10.3791/61443.
73. Eicher, A.K.; Kechele, D.O.; Sundaram, N.; Berns, H.M.; Poling, H.M.; Haines, L.E.; Sanchez, J.G.; Kishimoto, K.; Krishnamurthy, M.; Han, L.; et al. Functional human gastrointestinal organoids can be engineered from three primary germ layers derived separately from pluripotent stem cells. Cell stem cell 2022, 29, 36-51.e36, doi:10.1016/j.stem.2021.10.010.
110. Miao, X.; Wang, C.; Chai, C.; Tang, H.; Hu, J.; Zhao, Z.; Luo, W.; Zhang, H.; Zhu, K.; Zhou, W.; et al. Establishment of gastric cancer organoid and its application in individualized therapy. Oncology letters 2022, 24, 447, doi:10.3892/ol.2022.13567.
69. Li, G.; Ma, S.; Wu, Q.; Kong, D.; Yang, Z.; Gu, Z.; Feng, L.; Zhang, K.; Cheng, S.; Tian, Y.; et al. Establishment of gastric signet ring cell carcinoma organoid for the therapeutic drug testing. Cell death discovery 2022, 8, 6, doi:10.1038/s41420-021-00803-7.
111. Yoon, C.; Lu, J.; Kim, B.J.; Cho, S.J.; Kim, J.H.; Moy, R.H.; Ryeom, S.W.; Yoon, S.S. Patient-Derived Organoids from Locally Advanced Gastric Adenocarcinomas Can Predict Resistance to Neoadjuvant Chemotherapy. Journal of gastrointestinal surgery : official journal of the Society for Surgery of the Alimentary Tract 2023, 27, 666-676, doi:10.1007/s11605-022-05568-7.
90. Zhang, H.; Qin, Y.; Jia, M.; Li, L.; Zhang, W.; Li, L.; Zhang, Z.; Liu, Y. A gastric cancer patient-derived three-dimensional cell spheroid culture model. American journal of cancer research 2023, 13, 964-975.
Reviewer 5 Report
I thank the authors for the efforts in trying to modify the manuscript that is now improved.
Author Response

(The authors gave the same response as above.)
